# Genetically encoded transcriptional plasticity underlies stress adaptation in *Mycobacterium tuberculosis*

Cheng Bei[1,7], Junhao Zhu [2,3,7], Peter H. Culviner[2], Mingyu Gan[4], Eric J. Rubin [2], Sarah M. Fortune [2], Qian Gao [1,5] ✉ & Qingyun Liu [2,6] ✉

Transcriptional regulation is a critical adaptive mechanism that allows bacteria to respond to changing environments, yet the concept of transcriptional plasticity (TP) – the variability of gene expression in response to environmental changes – remains largely unexplored. In this study, we investigate the genome-wide TP profiles of *Mycobacterium tuberculosis* (*Mtb*) genes by analyzing 894 RNA sequencing samples derived from 73 different environmental conditions. Our data reveal that *Mtb* genes exhibit significant TP variation that correlates with gene function and gene essentiality. We also find that critical genetic features, such as gene length, GC content, and operon size independently impose constraints on TP, beyond trans-regulation. By extending our analysis to include two other *Mycobacterium* species -- *M. smegmatis* and *M. abscessu*s -- we demonstrate a striking conservation of the TP landscape. This study provides a comprehensive understanding of the TP exhibited by mycobacteria genes, shedding light on this significant, yet understudied, genetic feature encoded in bacterial genomes.

Cells must swiftly modulate the expression of their genes to cope with abrupt changes in the external environment. Transcriptional plasticity (TP)[1], which is defined as the ability to alter the expression of a gene in response to different types of environmental stress, is pivotal to cellular adaptation and subject to natural selection[2–4]. In practice, TP can be estimated by quantifying the change in the level of expression across multiple conditions. For instance, a gene with high TP exhibits substantial changes in expression across different conditions, while a gene with low TP maintains relatively stable expression regardless of environmental changes (Fig. 1a). Urchueguía et al. used a library of *E. coli* strains containing promoter-green fluorescence protein (GFP) fusions to measure changes in fluorescence levels across different

conditions, thereby quantifying expression plasticity[4]. Similarly, Lehner et al. used the normalized sum of squares of log2- expression changes to infer gene-level transcriptional plasticity from *Saccharomyces cerevisiae* microarray dataset[5]. These studies found that certain genetic traits, such as promoter architecture, nucleosome organization, and histone modification patterns, may significantly influence eukaryotic gene transcriptional plasticity[6–10]. While the transcriptional machinery and the nucleoid organization of prokaryotic organisms fundamentally differ from those of eukaryotes[11,12], a recent investigation into *E. coli* promoter evolution showed that long-term natural selection favors the retention of high promoter TP despite the presence of segregating mutations[2]. The strong evolutionary constraint

[1]Key Laboratory of Medical Molecular Virology (MOE/NHC/CAMS), School of Basic Medical Science, Shanghai Medical College, Shanghai Institute of Infectious Disease and Biosecurity, Fudan University, Shanghai, China. [2]Department of Immunology and Infectious Diseases, Harvard T. H. Chan School of Public Health, Boston, MA, USA. [3]CAS Key Laboratory of Pathogen Microbiology and Immunology, Institute of Microbiology, Chinese Academy of Sciences, Beijing, China. [4]Center for Molecular Medicine, Children's Hospital of Fudan University, National Children's Medical Center, 201102 Shanghai, China. [5]National Clinical Research Center for Infectious Diseases, Shenzhen Third People's Hospital, Shenzhen, Guangdong Province, China. [6]Present address: Department of Genetics, University of North Carolina at Chapel Hill, Chapel Hill, NC 27599, USA. [7]These authors contributed equally: Cheng Bei, Junhao Zhu. ✉e-mail: qiangao@fudan.edu.cn; qingyun_liu@med.unc.edu

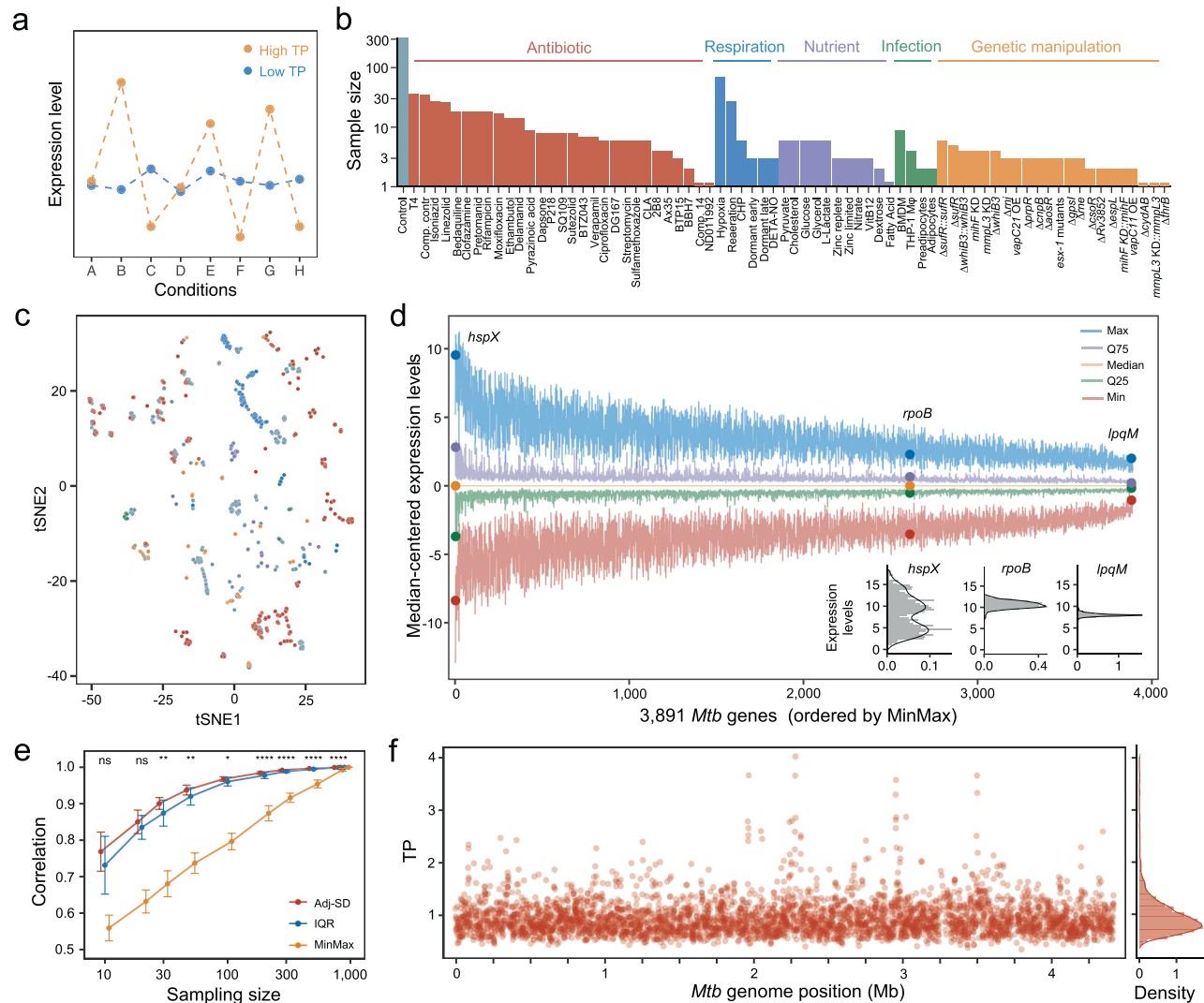

**Fig. 1 | Genome-wide estimation of *Mtb* transcriptional plasticity (TP). a** A schematic diagram of TP. **b** A diagram illustrating the composition of the 894 samples from 73 different conditions. Detailed information about the samples can be found in Supplementary Data 1. **c** Visualization of the 894 samples using t-distributed stochastic neighbor embedding (tSNE) grouped according to different experimental condition categories. **d** Primary expression statistics of *Mtb* genes across the 894 samples. The *X*-axis represents the ranking of 3891 *Mtb* genes ordered by their expression ranges (*MinMax*). The five line-plots represent the maximum (Max), 75 percentile (Q75), median, 25 percentile (Q25), and minimum (Min) expression levels which are centered by subtracting the median expression level of each gene. Expression statistics for three representative genes, *hspX*, *rpoB*, and *lpqM*, are highlighted. **e** Comparing adj-SD, IQR, and MinMax metrics in describing TP of *Mtb* genes using a subsampling and bootstrap analysis (see the "Methods" section). A subset of $N = 10, 20, 30, 50, 100, 200, 300, 500$, or 800 samples were randomly drawn from the full dataset. Statistical significance between correlation coefficients of adj-SD and IQR was estimated by Wilcoxon tests (two-sided), the corresponding *p* values were 0.096 ($N = 10$), 0.068 ($N = 20$), 0.001 ($N = 30$), 0.002 ($N = 50$), 0.010 ($N = 100$), $3.506 \times 10^{-5}$ ($N = 200$), $2.239 \times 10^{-6}$ ($N = 300$), $3.773 \times 10^{-6}$ ($N = 500$) and $4.109 \times 10^{-6}$ ($N = 800$). ns represents non-significant, *p value 0.01–0.05, **p value 0.001–0.01, ***p value 0.0001–0.001, and ****p value < 0.0001. Error bars represent the mean ± SD of TPs. **f** Genome-wide TP profiles (adj-SD) of the 3891 *Mtb* genes. The positively skewed genome-wide TP distribution is illustrated in the right panel.

implies that akin to eukaryotes, there may also be genetic traits in bacteria that determine TP, but the biological principles underlying TP in bacteria have not been adequately studied[4,13].

Exploring the genetic features contributing to TP in bacteria can enhance our understanding of how bacteria adapt to environmental pressures and guide the development of innovative strategies to combat bacterial pathogens. Tuberculosis (TB) remains the leading cause of death due to a single infectious agent[14]. Throughout the phases of infection, proliferation, and transmission, *Mycobacterium tuberculosis* (*Mtb*), the causative agent of TB, faces a wide array of environmental challenges. Some of the stresses, such as hypoxia, are characteristic of the microenvironments where the bacilli reside within the host, whereas others arise from host immune defenses such as toxic metal ions, nutrient restriction, acidic pH, and reactive

oxygen or nitrogen species, etc. Over the past 75 years, *Mtb* has also faced constant pressure from antibiotics. To understand how *Mtb* modulates its gene expression in response to different external challenges, studies have leveraged RNA sequencing (RNA-Seq) to query *Mtb*'s transcriptomic profiles across a broad panel of environmental conditions. These studies have revealed a complex transcriptional regulation network underlying the ability of *Mtb* to adapt to stresses. For example, over 50 transcriptional factors (such as *dosR* and *whiB3*) respond to hypoxia, allowing *Mtb*, an obligate aerobe, to survive in settings with oxygen depletion[15]. As these studies have been conducted under a multitude of experimental conditions, the resultant RNA-Seq datasets provide a comprehensive view of gene expression in *Mtb* that can be analyzed for insights into its transcriptional plasticity.

In this work, we systematically examine the TP profiles of *Mtb* genes by integrating publicly available RNA-Seq datasets. Our analysis uncovers significant variability in TP across genes and identifies overarching principles governing the amplitude of TP. We find a correlation between a gene's biological function and its TP and note that essential genes exhibit significantly lower levels of TP than non-essential genes. We further demonstrate that in addition to transcriptional regulators, genetic features such as operon architecture, gene length, and GC content (GC%) also appear to play substantial and distinct roles in shaping the TP of *Mtb* genes. In addition, by extending our study to *M. smegmatis* and *M. abscessus*, we show that the same principles appear to govern TP in other Mycobacteria. The findings in this study enrich our understanding of TP regulation and underscore the shared regulatory mechanisms governing gene expression dynamics.

## Results

### Quantifying the transcriptional plasticity of *Mtb* genes

To explore the transcriptome-wide pattern of gene expression in *Mtb*, we collected 894 previously published *Mtb* RNA-Seq samples that were generated under a wide range of experimental conditions. All of the 894 samples were obtained by studying the standard laboratory strain *Mtb* H37Rv, thus interrogating the physiological responses to various challenges in the same genetic background. These studies included antibiotic exposures, varied nutrient sources, host-mimicking conditions, and genetic manipulations such as gene knock-downs or deletions, as well as the corresponding untreated controls (Fig. 1b, see also the "Methods" section and Supplementary Data 1). We reasoned that the wide diversity of these experimental conditions would provide a suitable resource for studying the transcriptional plasticity of *Mtb* genes (Fig. 1c).

We first employed standardized preprocessing criteria to facilitate the analysis of the 894 RNAseq samples (see the "Methods" section). In brief, we excluded genes shorter than 150 bp, non-coding transcripts, and genes whose expression was not detected in most samples. We then normalized the expression data for the remaining 3891 genes using the trimmed mean of M-values (TMM) method, a technique designed to account for varying sequencing depth and suppress batch effect[16]. As expected, TMM normalization effectively adjusted the 894 datasets to have comparable expression medians (Fig. S1a) and reduced variations across batches (Fig. S1b–f). For subsequent TP analysis, the expression levels were indicated using TMM-normalized, log2-transformed Reads Per kilobase million (RPKM + 1) (Supplementary Data 2).

To estimate variations in gene expression, we initially calculated the range of expression levels, or the *MinMax*, of the *Mtb* genes across the 894 samples. We noticed that the *MinMax* of *Mtb* genes varied from 2.59 to 18.11 (Fig. S1g), suggesting that the amplitude of the changes in the level of expression for certain *Mtb* genes could exceed the range of expression of other genes by a factor of more than 40,000. We then examined gene expression at different percentiles of expression, including the most highly expressed 100th percentile (Max), the 75th (Q75), 50th (Median), 25th (Q25), and 1st (Min) percentiles—and observed significant differences in ranges of expression among *Mtb* genes (Fig. 1d). For instance, *hspX*—encoding a hypoxia-induced small heat shock protein[17]—displayed a markedly broader range of expression compared to *rpoB*, which encodes the β subunit of the RNA polymerase core enzyme. Conversely, the expression level of the lipoprotein peptidase gene, *lpqM*, remained almost constant across all conditions (Fig. 1d).

We further characterized variations in expression with two additional metrics: the Inter-Quantile-Range (IQR) and the mean-adjusted Standard Deviation (adj-SD) of the expression levels (Fig. S1h, see the "Methods" section). As expected, we found significant correlations between MinMax, IQR, and adj-SD with different degrees of correlation coefficients (Fig. S1i), indicating that these measures all represent the

variability of gene expression. To evaluate the robustness of these metrics, we performed a bootstrap analysis by comparing random subsamples with the complete dataset (see the "Methods" section). This analysis indicated that while both IQR and adj-SD were more resilient to reductions in sample size than MinMax (Fig. 1e), adj-SD demonstrated a slight but statistically significant advantage over IQR (Fig. 1e). Therefore, adj-SD was used to estimate TP in the subsequent analyses, and we calculated TPs for 3891 *Mtb* genes (Fig. 1f, Supplementary Data 3).

Because other technical factors could potentially affect the estimation of TPs, such as read coverage uniformity and GC content-associated sequencing bias[18], we conducted constrained analyses to assess their influence by controlling for each factor (Fig. S2a, b). TPs calculated from subsets of samples grouped by their degrees of mRNA coverage uniformity or GC% preference were still highly correlated with TPs calculated from the entire dataset (Fig. S2a, b). Additionally, to exclude the possibility that our results were biased by technical factors associated with experimental batches, we applied our analyses to three independent BioProjects with relatively large numbers of samples and estimated TPs within each of these batches (Fig. S2c). We found that TPs calculated from individual batches, despite their smaller sample sizes, still showed a high correlation with the TPs calculated from the entire dataset (Fig. S2d). Together, these analyses give us confidence that the TP estimation is robust to technical biases associated with different experimental batches.

### TP varies with gene function and gene essentiality

We found that TPs of 3891 *Mtb* genes displayed a predominantly normal distribution with a long tail representing genes with high TPs (Fig. 1f). Using a bootstrap approach similar to that described in Fig. 1e, we found that the 195 high-TP genes in the top 5% percentile demonstrated consistently high TPs even when the sample size was reduced to include just 10 samples (Fig. S2e, f). This pattern suggested that the skewed distribution was not caused by "outlier" values but instead reflected a subset of genes with a wider range of expression levels. Interestingly, there were a few "spikes" in the *Mtb* genome that exhibited extremely high TP (Fig. 1f), and most of these genes belonged to the DosR regulon (Fig. S2g). We then investigated the biological functions of the high-TP genes and found that the 195 high-TP genes were significantly enriched for genes involved in responding to stresses, including hypoxia, host immune mechanisms, copper ions, etc., as per the DAVID database[19] (Fig. 2a, Supplementary Data 4), and the genes within each functional group exhibited similar expression patterns (Fig. S3a, b). When we grouped *Mtb* genes based on previously established functional categories and compared their TP profiles[20,21], we found that genes involved in biomass production, cell wall biosynthesis, cellular metabolism, and respiration were primarily associated with the lowest TPs (Fig. 2b). This association is underscored by our observation that 1049 core genes, referring to genes conserved across mycobacteria species[22], exhibited significantly lower TPs compared to the other 2842 genes in the genome (Fig. 2c).

The above findings suggested that those genes crucial for basic cellular activities exhibit more tightly regulated expression. To test this, we compared the TP distribution for genes previously annotated as essential with those annotated as non-essential[23] and found that essential genes displayed significantly lower TPs than non-essential genes (Fig. 2d). We also noticed that those genes whose disruption by transposon insertion conferred a growth advantage exhibited significantly higher TPs than both essential and other non-essential genes (Fig. 2d). Recent studies have proposed gene vulnerability—the organism's susceptibility to perturbations in the transcription of the gene (e.g., by CRISPRi)—as a quantitative, orthogonal proxy to gene essentiality[24]. Consistent with the analysis by annotated essentiality, we found that genes identified as vulnerable also tended to exhibit lower *TP*s (Fig. 2e), and none of the highly vulnerable genes exhibited

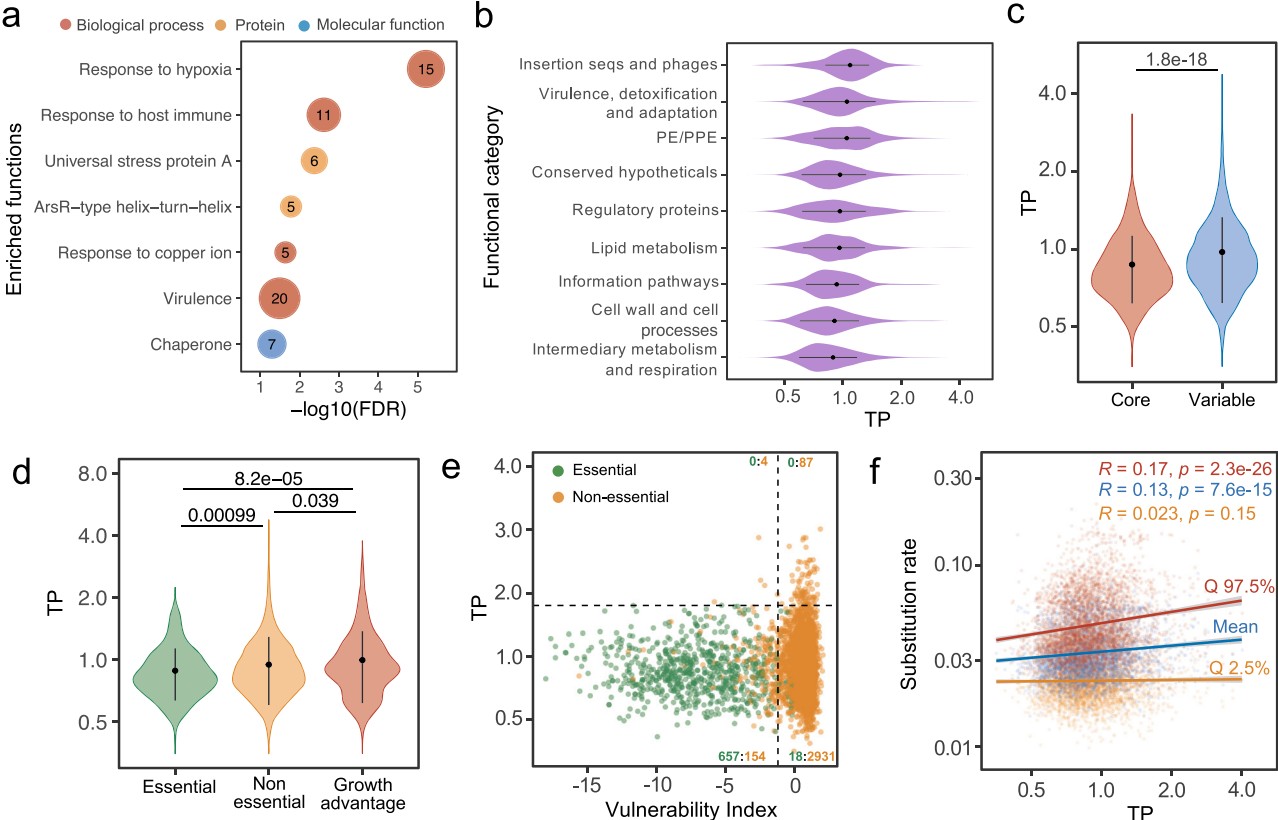

**Fig. 2 | TP is associated with gene function and gene essentiality. a** Functional enrichment analysis of the 195 high-TP genes. Numbers in the dots represent the number of genes in each category. **b** Violin plots showing the TP profiles of genes in different functional categories, where "Insertion seqs and phages" has 73 genes, "Virulence, detoxification and adaptation" has 236 genes, "PE/PPE" has 160 genes, "Conserved hypotheticals" has 1007 genes, "Regulatory proteins" has 197 genes, "Lipid metabolism" has 268 genes, "Information pathways" has 238 genes, "Cell wall and cell processes" has 762 genes, and "Intermediary metabolism and respiration" has 918 genes. Error bars denote mean ± SD of TPs. The *X*-axis is presented on a log scale. **c** 1049 genes of the mycobacterial core-genome exhibit lower TPs than the other 2842 genes of the variable genome. Error bars represent mean ± SD of TPs. Statistical significance was assessed by the Wilcoxon test (two-sided). **d** TP comparison between 459 essential genes, 2874 non-essential genes, and 301 genes whose disruption confers growth advantage under axenic culture conditions. Statistical significance was assessed by the Wilcoxon test (two-sided); error bars represent mean ± SD of TPs. **e** *Mtb* Genes vulnerable to transcriptional perturbation exhibit low TPs. The horizontal black dashed line represents the maximum TP value of essential genes, and the vertical line shows the 5th percentile of vulnerability index of non-essential genes. The counts of essential and non-essential genes in each quadrant are displayed in green and yellow, respectively. **f** TP positively correlates with genes' substitution rate, as simulated by *genomegaMap* (Wilson, 2020). Mean value and 95% credibility intervals of substitution rates are presented in colored points. Colored Lines depict the linear fit between TP and substitution rate. *R* and *p* represent Spearman's correlation coefficient and the associated *p* values, respectively.

high TP (Fig. 2e). We hypothesized that high-TP genes may promote phenotypic diversification that confers a selective advantage in the ability of *Mtb* to survive in fluctuating environments, and therefore these genes might exhibit rates of evolution that differ from the rest of the genome. To test this hypothesis, we utilized a recently established set of evolutionary metrics for *Mtb* genes drawn from 10,209 *Mtb* genomes[25]. We found that high-TP genes exhibited higher base substitution rates than low-TP genes (Fig. 2f). Overall, our analysis suggests that for those genes involved in essential cellular processes, stable levels of expression are advantageous to the bacteria. In contrast, for genes that provide a growth advantage in certain conditions but are dispensable or even detrimental in others, a "plastic" inducible transcriptional program appears to be beneficial.

### Genetic features underlying transcriptional plasticity

To identify the genetic factors influencing TPs of *Mtb* genes, we compiled a comprehensive list of 119 genetic features, including promoter and CDS sequence composition, transcriptional regulation, evolutionary parameters, and gene vulnerability (Fig. 3a, Supplementary Data 5, see the "Methods" section). We then employed a decision-tree-based regression analysis (Light Gradient-Boosting Machine) to model the *Mtb* TP landscape with these 119 features (Fig. 3b). The regression model

was trained on a randomly selected subset of 60% (2335/3891) of the total *Mtb* genes, and then used to predict the TPs of the remaining 40% (1556/3891) of *Mtb* genes. We iterated this process 100 times, with the derived models yielding an average $R^2$ value of 0.25 (Fig. S4a). For each model, the features were ranked by importance based on the contribution of each feature to the predictive power of the model. We then aggregated these feature ranks across all iterations to provide an average measure of each feature's contribution to TP prediction.

Our analysis highlighted four features—operon length, gene length, number of activating regulators, and GC percentage (GC%)—that consistently demonstrated high predictive importance across iterations (Fig. 3c). A support vector machine (SVM) model trained solely with these four features was able to predict a gene's TP ($R^2 = 0.18$ with an accuracy slightly lower than that of a model trained with all 119 features (Figs. 3d and S4b). We observed no or low-level correlations between the top features (Fig. S4c).

### The role of genetic features in affecting transcriptional plasticity

We then sought to understand how these features influence TP. We first examined the role of gene length and found a negative correlation between gene length and TP, with longer genes tending to exhibit lower TPs than shorter genes (Fig. 4a). Contrary to gene length;

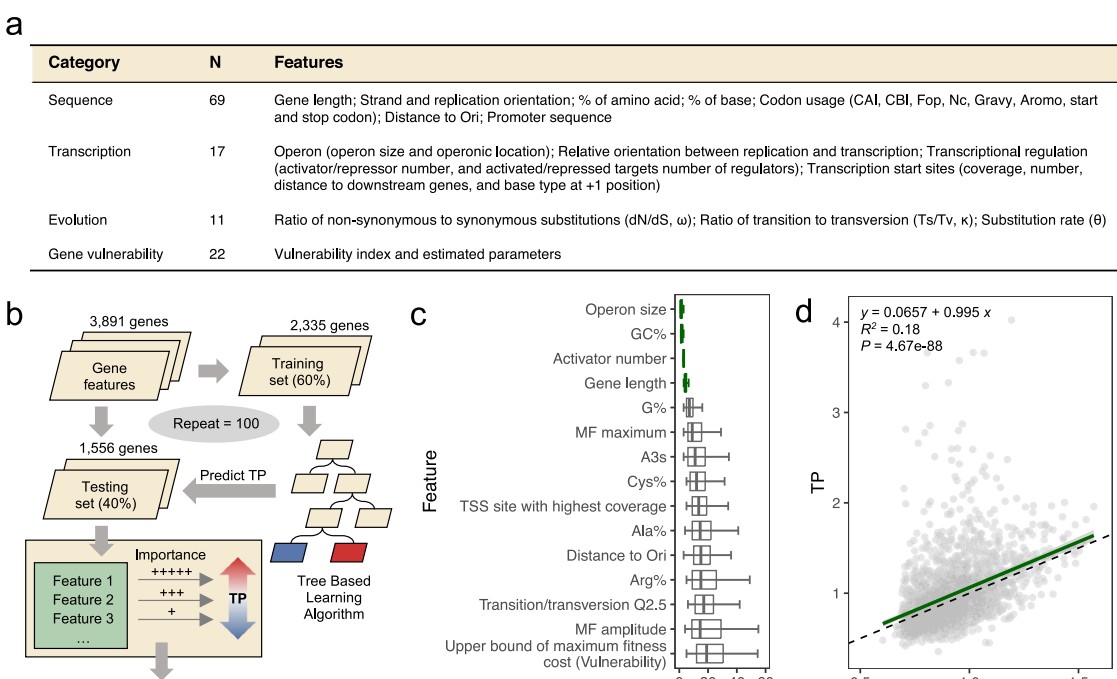

**Fig. 3 | Identification of genetic features underlying TP. a** A table summary of the 119 candidate genetic features. *N* denotes the number of features in each category. **b** Schematic diagram illustrating our machine-learning workflow. **c** The top 15 genetic features ranked by their median feature importance in predicting TP. Lower ranks signify higher feature importance for TP prediction, whereas a tight rank distribution indicates higher consistency in predictions across randomized sample splits and modeling iterations. The four genetic features that consistently rank low across random repeats are highlighted in green. Boxes show the median, the 1st and 3rd quartile of feature importance ranks (*N* = 100) across experiments, and the whiskers represent the median ± 1.5 × IQR (interquartile range). Vertical lines in boxes represent the medians. **d** An SVM model constructed using only the top four features effectively predicts TP. The green line represents the linear fit between SVM-modeled and observed TPs. The black dashed line represents the formula *y* = *x*. Error band represents the 95% confidence interval. Pearson's correlation coefficients and the corresponding *p* values are presented.

however, the correlation between TP and GC content (GC%) was not monotonic. We found that genes with a GC% substantially different from the genome-wide GC% of *Mtb* (65.6%) generally had higher TPs (Fig. 4b). To confirm this observation, we binned *Mtb* genes according to their TPs and calculated the standard deviation (SD) for GC% of the genes in each bin. We observed an apparent linear correlation between the SD of the GC% and the ranks of TP bins, such that the bins with higher TPs had larger SDs for GC% (Fig. 4c). We also calculated gene-level GC% deviation from the genome-wide GC% and observed a positive correlation between GC% deviation and TP (Fig. S5a). This corroborated the hypothesis that the TP increases with greater GC% deviation from the genome-wide GC%. Notably, both essential and non-essential genes whose GC% approximated the genome-wide GC% exhibited lower TPs (Fig. S5b, c), implying that the association between GC% and TP was not confounded by gene essentiality.

Next, we evaluated the effect of operon size on TP. We found that genes located in polygenic operons, containing two or more genes, had significantly higher TPs than genes located in monogenic operons, consisting of only one gene (Fig. 4d). Furthermore, we also observed that the TPs of genes within the same operon were highly correlated (Fig. S5d) and the similarity of TPs and expression profiles between operonic genes negatively correlated with the number of other genes in the operon between the genes being examined (Figs. 4e and S5e). Despite the confounding TP differences between essential and non-essential genes, both exhibited higher TPs in polygenic operons (Fig. S5f, g). A recent study reported that *Mtb* undergoes frequent premature transcription termination[26]; while we did observe a decreased mean expression for downstream genes in an operon (Fig. S5h), there was no such trend for TP (Fig. S5i). Together, these analyses implied that it is the size of the operon, rather than the position of the genes within the operon, that influences TP.

While gene length, GC%, and operon size are features related to the primary sequence of the gene, the number of activating regulators is a feature that pertains to the process of transcriptional regulation. We found that the TP of a gene tended to be higher when its expression was modulated by a higher number of predicted transcriptional activators (Fig. 4f). We also observed a similar trend for transcriptional repressors, whereby genes with more predicted repressors tended to have higher TPs, although the TP dropped slightly in genes predicted to have only one repressor (Fig. 4g). Taken together, our analysis shows that not only the basic genetic composition of genes but also the complex network of transcriptional regulation can significantly influence the TP landscape of the *Mtb* genome (Fig. 4h).

## Genetic features can explain TP variation in genes belonging to the same regulon
Despite the extraordinary complexity of transcription regulation, recent studies suggested that bacterial genes can be roughly grouped into clusters, or "regulons", based on concordant expression patterns across conditions. We, therefore, speculated that for genes of the same "regulon", they might also exhibit similar TPs. We investigated 36 well-annotated gene regulons (see the "Methods" section, Supplementary Data 6) and found that the TP varied greatly between different regulons (Fig. 5a). For instance, the regulons "Mce3R" and "KstR2", which are thought to be involved in lipid metabolism[27,28], had lowest TPs (Fig. 5a). By contrast, the hypoxia- and redox-sensing "DosR" regulon and metal related regulons such as "Zur", "RicR" and "IdeR", demonstrated high TPs (Fig. 5a). However, while the genes within the same regulon displayed similar expression patterns—expression levels went up or down in a coordinated manner—they differed significantly in the magnitude of their changes in expression, resulting in diverse TPs. For example, the TPs of the genes belonging to the "DosR" regulon varied substantially,

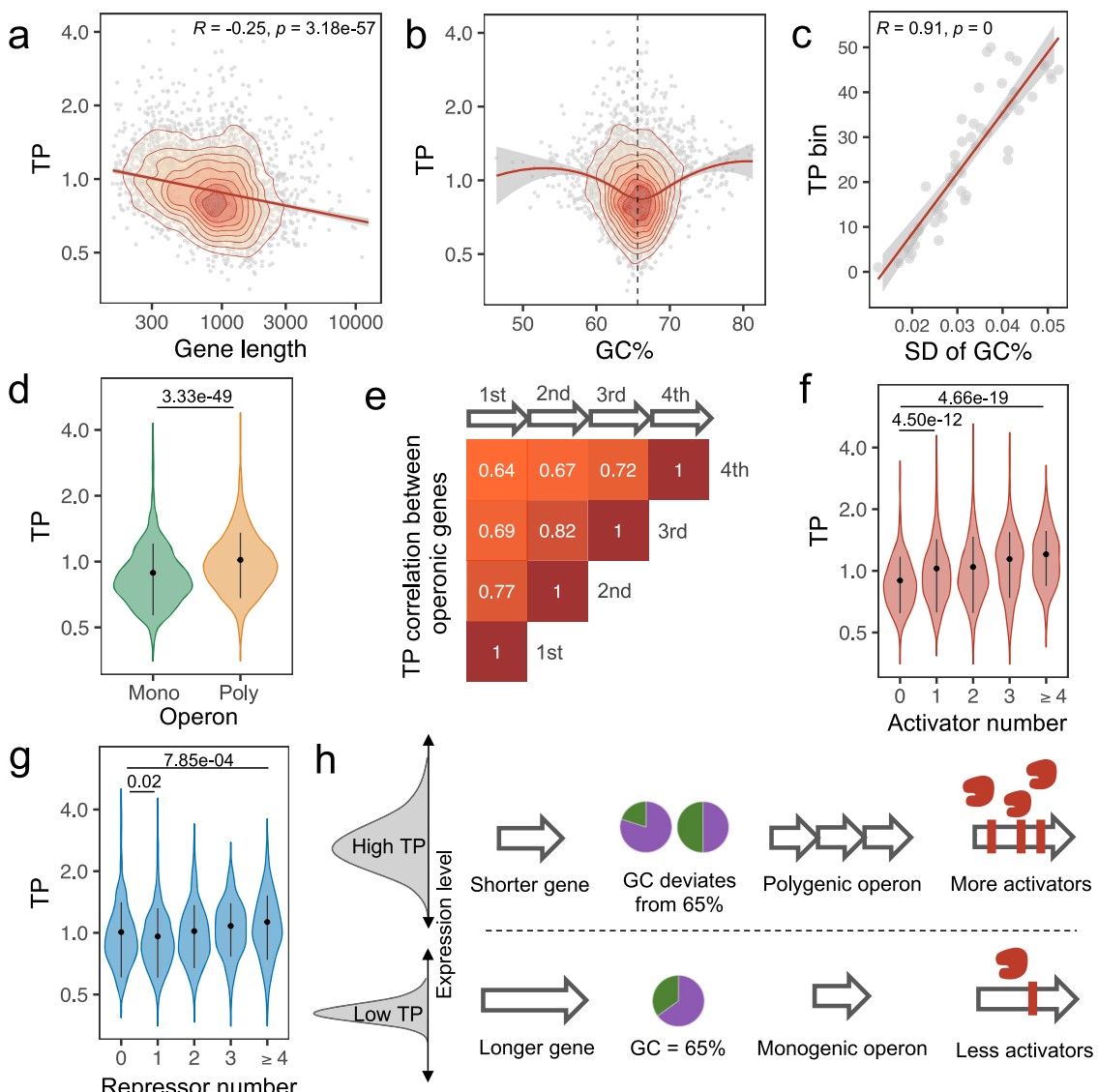

**Fig. 4 | Impact of key genetic features on TP. a** A negative correlation exists between gene length and TP, illustrated by the 2D density contour plot of genes by TP and gene length. The red line depicts the linear fit. Error band represents the 95% confidence interval. *R* and *p* represent Spearman's correlation coefficient and the associated *p* values, respectively. **b** Deviation in GC% from the genome-wide GC% (65.6%, black dashed line) is positively linked with TP, depicted by the LOESS trendline and the 2D density contours. Error band represents the 95% confidence interval. **c** Positive association between average TP bin (divided into 50 TP bins) and standard deviation (SD) of GC% in each bin. Error band represents the 95% confidence interval. *R* and *p* represent Spearman's correlation coefficient and the associated *p* values, respectively. **d** 1567 genes in polygenic operons exhibit

significantly higher TPs than 2235 genes in monogenic operons. Wilcoxon tests (two-sided). Error bars represent mean ± SD of TPs. **e** Pearson's correlation coefficients of TP between genes in different operonic positions, i.e., the first, the second, the third, and the fourth gene of an operon. **f** and **g** TP increases as genes are regulated by more regulators. Boxplots demonstrate a monotonic relationship between TP and the number of activators. Genes targeted by only one repressor display the lowest TPs. Numbers of genes targeted by 0, 1, 2, 3, and ≥4 activators are 743, 791, 285, 117, and 104, and the numbers of genes targeted by 0, 1, 2, 3, and ≥4 repressors are 637, 916, 288, 116, and 83. Error bars represent the mean ± SD of TPs. Statistical significance was assessed by Wilcoxon tests (two-sided). **h** A schematic illustrating the relationships between the four genetic features and TP.

with *dosT* exhibiting the lowest (TP = 0.73) and *hspX* exhibiting the greatest change in level of expression (TP = 4.02) (Fig. 5b, c). Similar TP variations were observed amongst the genes belonging to other regulons (Fig. 5a). Because the expression of genes within a regulon generally showed the same direction of change in response to stress, we speculated that the TP differences amongst the regulon's genes might derive from differences in the genetic features of the individual genes. Indeed, we found the two primary genetic features—gene length and GC %—could partly explain the TP variations of co-regulated genes in most regulons (Fig. S6, 7). To show this, we selected five regulons that comprised of more than 20 genes each ("WhiB1", "WhiB4", "Zur", "DosR", and "Rv1828/SigH") and demonstrated that shorter genes with a

GC% deviating from the genomic-wide GC% generally displayed higher TP than other co-regulated genes (Fig. 5d, e). These results highlight the ability of genetic features to affect the TP, independent of other transcriptional regulatory processes.

## The transcriptional plasticity landscape is conserved across *Mycobacterium* species

The analyses above revealed that in *Mtb*, a gene's TP is linked to its function, essentiality, and evolutionary and genetic features, all of which are likely to be conserved in closely related homologous genes from other mycobacterial species. To demonstrate this, we curated published RNA-Seq datasets from 192 samples of *M. smegmatis* (*Msm*)

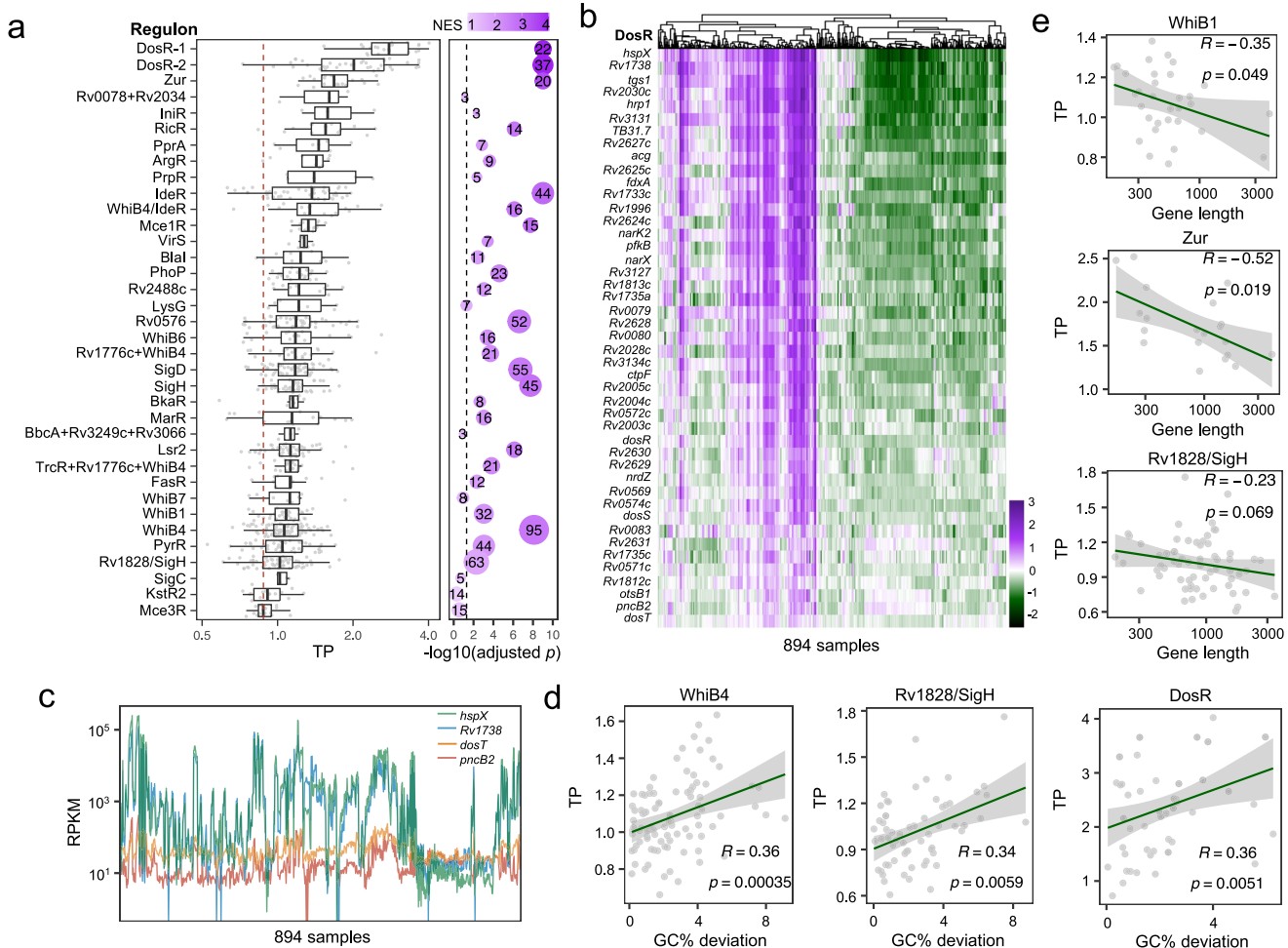

**Fig. 5 | The impact of primary sequence features on TP is partially independent of transcription regulation. a** *Mtb* regulons display varying degrees of transcriptional plasticity. Boxes show the median, the 1st and 3rd quartile of TP, and the whiskers represent the median ± 1.5 × IQR. Vertical lines in boxes represent the median. The red dashed line represents the median TP of all 3891 genes. The bubble plot to the right summarizes the statistical significance (adjusted *p*-value) and normalized enrichment score (NES) of each regulon by single-sample gene set enrichment analysis (ssGSEA). A higher NES indicates that the operon is enriched for genes with higher TPs. Bubble size corresponds to the number of genes in each regulon. Numbers in the dots represent the number of genes in each regulon. One-sided adjusted *p*-value was calculated for each regulon. **b** Expression profiles of DosR regulon genes ranked by TP. The color gradient represents the *Z*-score

normalized log-RPKM. **c** Variations in TP within the DosR regulon, exemplified by comparing expression profiles of two high-TP genes (*hspX* and *Rv1738*) with two low-TP genes (*dosT* and *pncB2*). **d** Deviation in GC% from the genome average partially explains TP variations of genes of the same regulon. Linear fits, Spearman's correlation coefficients, and the corresponding *p* values are shown for three representative regulons, WhiB4, Rv1828/SigH, and DosR. Error bands represent the 95% confidence interval. **e** TPs of co-regulated genes negatively correlate with their gene lengths. Spearman's correction coefficient and the corresponding *p* values are provided. The associations between primary genetic features and TP for genes in additional regulons are illustrated in Figs. S6, 7. Error bands represent the 95% confidence interval.

and 106 samples of *M. abscessus* (*Mab*) and used *adj-SD* to estimate their genome-wide TP (Fig. S8a, Supplementary Data 7). We found that all three species displayed comparable TP distributions, which were positively skewed with long tails harboring high-TP genes (Fig. 1f, Fig. S8b). A closer examination indicated that homologous genes among *Mtb*, *Msm*, and *Mab* exhibited similar amplitudes of TP (Figs. 6a, b, and S8c). Moreover, as observed in *Mtb*, the essential/vulnerable genes in *Msm* exhibited lower TPs than non-essential or less vulnerable genes (Fig. 6c, d). Also, as seen in *Mtb*, the genes in *Msm* and *Mab* with higher TP values tended to be shorter in length and have GC% more deviated from the genome-wide GC% (Fig. 6e–h). It is intriguing that the high-TP genes across all three species were enriched in metal-related functions (Figs. S8d, e, 3a). These findings suggest that despite the differences in natural lifestyles, the evolutionary principles underlying TP are likely conserved across mycobacterial species.

On the contrary, although the TP landscape overall demonstrated conservation across the three species, the outliers—genes exhibiting

distinct TPs in different *Mycobacterium* species—could be related to their niche adaptation. For instance, compared to *Msm* and *Mab*, we observed significantly higher TPs in genes related to amino acid biosynthesis in *Mtb* (Fig. S8f, g). Intriguingly, amino acid biosynthesis, such as arginine synthesis, has been found to be involved in *Mtb*'s responses to oxidative stress, DNA damage, and host immune pressure[29]. Therefore, the elevated TP levels of these genes may represent an adaptation to the host microenvironment."

## Discussion

In this work, we assessed the TP of *Mtb* genes by utilizing 894 RNA-Seq datasets that were previously collected when the bacteria were exposed to various environmental conditions. Our analyses revealed that TP varies significantly among *Mtb* genes in a manner that is associated with their biological functions and is subjected to natural selection. We identified primary genetic features that contribute to TP values, including gene length, GC%, operon size, and transcriptional

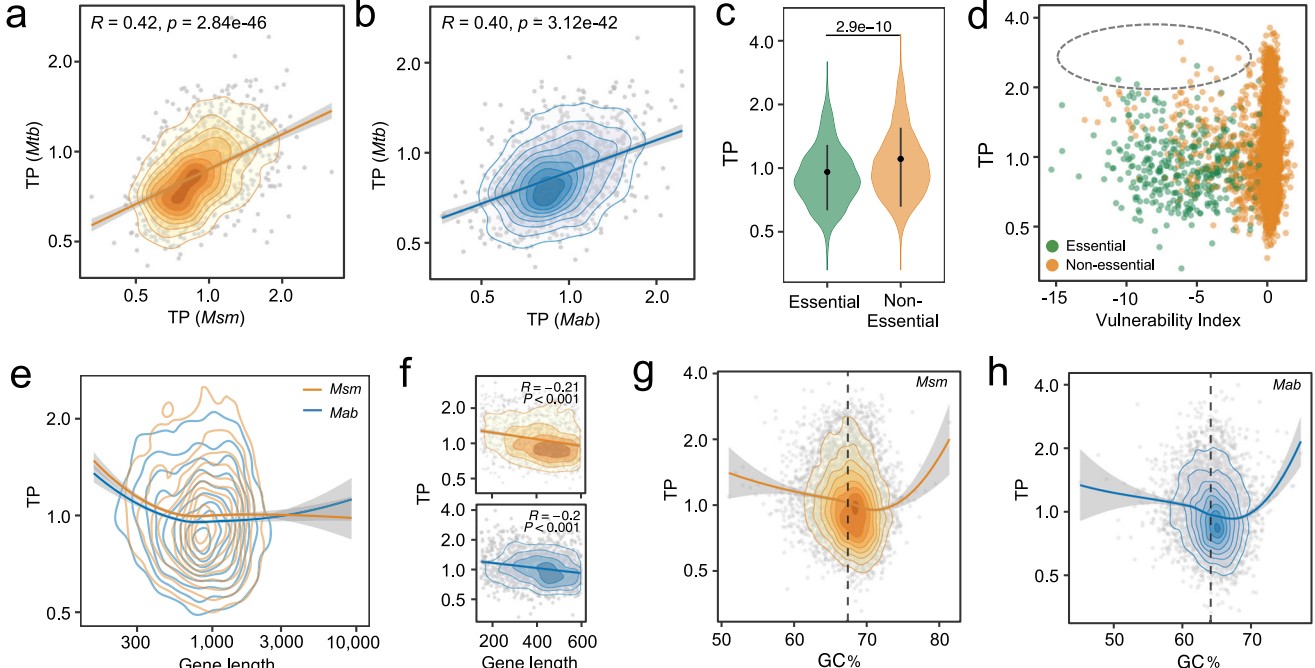

**Fig. 6 | TP and its underlying genetic determinants are conserved in other *Mycobacterium* species. a** and **b** The TP profiles of *M. smegmatis* (*Msm*) and *M. abscessus* (*Mab*) genes resemble those of the *Mtb* homologs. The 2D density contour plots illustrate the distribution of gene orthologs according to their TPs in corresponding *Mycobacterium* species. Lines denote the linear fits. Error bands represent the 95% confidence interval. Pearson's correction coefficient and the corresponding *p* values are provided. **c** 5875 non-essential *Msm* genes have higher TPs than 387 essential *Msm* genes. Error bars represent mean ± SD of TPs. Statistical significance was measured by Wilcoxon tests (two-sided), and the corresponding *p*-value was presented. **d** *Msm* genes vulnerable to transcriptional perturbation exhibit low TPs. The gray circle highlights the lack of genes with both high TP and

high vulnerability. **e** Gene length is negatively associated with TP in *Msm* (orange) and *Mab* (blue). The 2D density contour plots illustrate the distribution of genes based on TP and gene length. Error bands represent the 95% confidence interval. **f** A linear correlation is observed between TPs and gene lengths for genes shorter than 600 bp. Error bands represent the 95% confidence interval. Spearman's correction coefficient and the corresponding *p* values are provided ($p = 2.53e-17$ for *Msm* and $2.85e-14$ for *Mab*). **g** and **h** Genes with GC% close to the genome-wide GC% (67.4% in *Msm* and 64.1% in *Mab*, annotated by black dashed lines) display lower TP in both *Msm* (**g**) and *Mab* (**h**). The 2D density contour plots depict the distribution of genes by their TPs and GC%. Error bands represent the 95% confidence interval.

regulatory factors. Finally, we extended these findings to *Msm* and *Mab*, demonstrating that TP, and the factors that influence it, are likely to be biological features that are conserved across mycobacterial species.

Gene vulnerability reflects the quantitative association between changes in bacterial fitness and the degree of CRISPR-mediated inhibition of a gene's transcription[24]. Perturbing the expression of highly vulnerable genes can be deleterious, whereas the same level of expression inhibition of invulnerable genes can be tolerated[24]. Initially, we anticipated a linear-like relationship between TP and vulnerability, whereby more vulnerable genes would exhibit lower TPs. Although we observed a positive association between vulnerability and TP, this relationship could not be explained by a simple or log-linear model. Instead, we observed an intriguing pattern between vulnerability and TP that presented as a reversed "L"-shape, with the elbow point representing genes that were insensitive to transcription inhibition and invariant in expression. This observation could be due to several reasons. First, the effects on the bacteria caused by the gene's transcriptional activation or transcriptional repression are not necessarily symmetrical. For instance, for some house-keeping genes, over-expression is better tolerated by the bacteria than repressed expression, whereas for protein toxins, the outcomes would be the opposite[30]. Because TP considers both up and down-regulation of gene expression, it reflects gene-specific constraints on both transcriptional activation and repression, whereas studies of gene vulnerability and essentiality only consider transcriptional repression. Second, vulnerability is not a constant gene feature but rather is expected to vary depending on the specific environmental conditions. Therefore, we

speculate that vulnerability estimated from different conditions could have a stronger correlation with TP. Finally, although essential genes showed significantly lower TP than non-essential genes, the TP variation in essential genes is overall quite close to that of non-essential genes. This suggests that bacteria may have the flexibility to alter the level of expression of essential genes as required for survival in changing environments (Fig. 2e).

It is noteworthy that primary genetic features (e.g., gene length, GC%, operonic structure) play important roles in determining TP, even though the mechanisms underlying this observation are not yet fully understood. For example, we found that shorter genes had higher TPs, a pattern that has also been observed in eukaryotes such as *Drosophila* and *Arabidopsis thaliana*[31,32]. The length of the gene appears to be evolutionarily shaped to accommodate its functionality, with housekeeping genes tending to be longer while stress-responsive genes tend to be shorter[32–34]. We speculate that stress-responsive genes require efficient and diverse expression patterns to cope with fluctuating environments while conserving energy. A reduction in gene size may represent an adaptive strategy to achieve this efficiency, allowing for more efficient regulation of the expression of these genes in response to stress. However, further research is needed to test this hypothesis and fully understand the evolutionary relationship of gene size with stress response.

There was a significant association between gene expression patterns and GC content, indicating that GC content could be an important regulatory factor[35]. It was previously observed that AU-rich and GC-rich transcripts follow distinct decay pathways, with a linear relationship between higher GC content and greater RNA stability[36]. In

our study, however, we found a "V" shaped relationship, whereby genes with low TP were clustered around a GC content of 65.6%, which is the genome-wide GC% of the *Mtb* genome. This finding contradicted our initial assumption that higher GC content would be associated with lower TP. The genes with extremely high GC content (>75%) may result from recent horizontal gene transfer from other bacteria[37–39], and therefore one possible explanation is that the TP of these recently acquired genes has not yet been optimized to align with the local transcriptional network, resulting in noisy expression of these genes. Moreover, high GC content may have a detrimental effect on expression stability if it leads to the formation of secondary structures or interferes with the binding of regulatory factors. The clustering of low TP genes at the *Mtb* genome-wide GC% (65.6%) suggests that these genes have evolved to be both GC stable and expression stable, thereby representing an optimized state of gene regulation.

Though we successfully identified four significant contributing features, the models incorporating these features could not completely predict TP values, suggesting that there are likely other determinants that were not identified (Figs. 3d, S4a), such as the promoter. Recent work in *E. coli* showed that, for most genes, the range of protein abundance across different environmental conditions is constrained by the TFs that regulate promoter activity[40]. Another study revealed that promoter characteristics, such as the length of the transcriptional initiation region and the presence of TATA-boxes, play important roles in determining the range of expression variation in eukaryotic genes[31]. Similarly, the positive correlation we found between TP and the number of transcriptional activators demonstrates the influence of promoter characteristics and trans-regulatory mechanisms on TP in mycobacteria (Fig. 4f, g).

As TP provides an empirical reference for gene expression variability across different conditions, we reason that knowing a gene's TP could have value by suggesting its role in *Mtb* physiology. For example, while essential genes are generally associated with low TPs, there are some non-essential/invulnerable genes that also exhibit very low TPs. One intriguing example is *Rv0012*, which is highly expressed (median log2-RPKM > 7) but has a very low TP (0.47). *Rv0012* encodes a membrane protein that is conserved among mycobacterial species but is not essential for *Mtb*'s growth either in vitro or during infection[24,41]. However, a recent chemical-genetic screening revealed that transcriptional repression of Rv0012 substantially sensitized *Mtb* to multiple antibiotics, especially antibiotics that target cell wall biosynthesis, such as vancomycin[42]. While it remains unclear why the *Mtb* genome harbors a group of non-essential, low-TP genes, this example suggests that this unique gene subset warrants closer examination and demonstrates that the TP may be useful as a supplement to gene essentiality and vulnerability for guiding gene candidate prioritization.

On the other hand, the expression profiles of low-TP genes are not always invariable; the few conditions where their expressions exhibit changes could provide clues to their physiological function. In future differential expression analyses, significant changes in the expression of genes with low TPs should be given more attention, as deviations from their usual level of expression could implicate them in the bacteria's response to particular experimental settings. In contrast, caution should be exercised regarding the differential expression of high-TP genes, as they often exhibit changes in expression even in "control" samples subjected to "no treatment". For example, changes in the expression levels of genes in the DosR regulon are frequently observed in control samples, presumably due to subtle differences from one experiment to another in parameters such as oxygen concentration. Therefore, it is worthwhile checking the validity of high-TP values by looking at the consistency of expression amongst samples within a group, such as either treated or untreated.

The inherent differences in the TP of genes can be used to normalize expression differences in microbial transcriptional studies. Traditionally, different thresholds have been employed to identify meaningful changes in gene expression. The threshold for identifying genes that respond to particular conditions is often a 2-fold change in the level of expression, or occasionally thresholds of 1.5-fold or 4-fold are used, but the genes exhibiting the largest transcriptional changes frequently receive the most attention. However, these thresholds are arbitrary because they don't adjust for the inherent TP of each gene. As a result, high-TP genes are more likely to display changes in expression that surpass the threshold, while relatively large changes in the expression in low-TP genes may be overlooked because they don't meet the arbitrary threshold. An alternative method would determine the degree of expression change that should be considered meaningful for each specific gene. To this end, a "soft-thresholding" benchmark for screening differentially expressed genes can be used by utilizing the expression changes corresponding to the 5th and 95th percentiles in our dataset (Supplementary Data 8). For instance, in the case of low-TP genes such as *lpqM* and *ribF*, the log2 fold-changes corresponding to the 95th quantile expression levels were 0.76 and 0.53, respectively, times the level of expression in the controls. An analysis using the arbitrary thresholds would miss changes in the expression of these genes that are equivalent to two standard deviations. Criteria based on the inherent TP for each gene could establish a more nuanced analysis for identifying differentially expressed genes. We believe that our integration of RNA-Seq data from 894 *Mtb* samples provides a comprehensive estimation of the transcriptional variations in *Mtb* genes across various conditions, and therefore the calculated TPs can serve as a reference for evaluating changes in expression. The TMM method employed in our analysis can be used to evaluate the transcriptional signatures of genes of interest (Supplementary Data 2). This will foster a deeper understanding of the differential gene expression landscape in *Mtb* and facilitate the exploration of gene-specific transcriptional patterns.

Alternatively, we propose incorporating TP into RNA-Seq analysis as a normalization factor. In our curated dataset of 894 RNA-Seq samples, we observed that the fold changes of differentially expressed genes (DEGs) were positively correlated with their TP values (Fig. S9a, see the "Methods" section). A potential way to diminish this effect is to divide log2 fold change (logFC) by the gene's TP. By doing this, logFC values of low-TP genes would be divided by smaller numbers than high-TP genes, which would result in an increase in low-TP genes' ranking in DEGs. After this normalization, the positive correlation between "TP-adjusted logFC" and TP was reduced (Fig. S9b), while the values of logFC and TP-adjusted logFC were still highly correlated (Fig. S9c, Pearson's correlation coefficient: 0.93).

In summary, our study has characterized the landscape of TP in *Mtb* genes and established a framework for determining TP levels. This work thereby serves as a foundation for future investigation aimed at understanding the influences that determine a gene's TP. Additionally, the proposed TP-based benchmark offers valuable guidance for the interpretation of differential expression changes in transcriptional studies. Moving forward, further research can build upon these findings to uncover the intricacies of TP and its impact on gene expression in *Mtb* and other microbial systems.

## Methods
### Collection and processing of RNA-Seq data
We used the keyword "tuberculosis" to search for publicly available RNA-Seq data of *Mtb* released on NCBI sequence read archive (SRA) before January 1, 2022, and obtained a total of 1084 datasets from 64 BioProjects with 47 associated research articles (Supplementary Data 1). FASTQ files of all 1084 samples were downloaded using Fastq-dump (version 2.8.0). Adapter trimming and the removal of low-quality sequencing reads were conducted using Trimmomatic (version 0.39)[43] with parameters of "ILLUMINACLIP:TruSeq3-PE.fa:2:30:10 LEADING:3 TRAILING:3 SLIDINGWINDOW:4:15 MINLEN:36" for paired-end data and "ILLUMINACLIP:TruSeq3-SE:2:30:10 LEADING:3

TRAILING:3 SLIDINGWINDOW:4:15 MINLEN:36" for single-end data. The filtered profiles were then mapped against the H37Rv reference genome (ASM19595v2) using Bowtie2 (version 2.2.9)[44], and duplicated reads were removed with SAMtools (version 1.9)[45]. To measure the read coverage distribution of each sample, we used "geneBody_coverage.py" from RSeQC (version 3.0.1)[46] that scaled all transcripts to 100 nt and calculated the number of reads covering each nucleotide position. To measure read coverage uniformity, we took gene body coverage profile across the 100 interpolated positions for each sample and calculated its Coefficient of Variation.

To identify the strand specificity of the RNA-Seq libraries, we measured the Pearson correlation coefficient of total read counts on two strands for each library using SAM files generated by Bowtie2. Libraries with a correlation coefficient lower than or equal to 0 would be considered as strand-specific, while a coefficient higher than or equal to 0.6 would be considered as non-strand-specific. For libraries with coefficients between 0 and 0.6, we manually judged their strand specificities based on the description of the experimental design and strand specificities of other samples from the same experiment. Library read counts were then enumerated with htseq-count from the HTSeq framework (version 0.11.3)[47] using non-strand-specific or strand-specific parameters based on strand specificities identified above. Samples with a small library size (<1,000,000 reads) and from Mtb strains other than H37Rv were excluded. 894 samples from 58 BioProjects were eventually included for further analysis. RNA-Seq data of Msm (mc²155) and Mab were collected and processed with the same pipeline used for Mtb. Msm data were mapped to the mc²155 reference genome (ASM1334914v1), and Mab data were mapped to ASM402801v1. Included were 293 Msm samples from 36 BioProjects and 146 Mab samples from 9 BioProjects.

## Quantification of transcriptional expression

Before library normalization, we removed small genes (≤150 bp), non-coding transcripts (tRNA, rRNA, and annotated non-coding RNAs in the Mtb genome), as well as non-expressing genes (read counts in all samples were zero). Read counts from each BioProject were subsequently normalized to account for variations in library size using the trimmed mean of M-values (TMM) factor[16], and the TMM normalized RPKMs were calculated using the edgeR package (version 3.30.3)[48]. Global TMM normalization was applied to the entire dataset (894 samples) to account for cross-BioProject batch effect. After TMM normalization, $\log_2$ (RPKM + 1) was calculated and defined as transcriptional expression levels. The Shannon index (SI) was calculated for each gene using the diversity function from the R package vegan (version 2.5–7). We then excluded samples from all three mycobacteria with a high proportion of zero-expressing genes (>4% of total genes) and also excluded genes with low SI (SI < 6.5 in Mtb, <4 in Msm and Mab) and genes that are not expressed in more than 1% of total samples. Downstream analyses thus included curated transcriptomic profiles of 894 samples and 3,891 genes from Mtb, 192 samples and 6311 genes from Msm, 106 samples, and 4839 genes from Mab (Supplementary Data 1).

## Stress conditions of RNA-Seq samples

To investigate the diversity of selected samples, we generalized the conditions of 894 samples based on the description in each BioProject and the related research articles. We further divided these conditions into six groups to summarize the sample conditions (Fig. 1b, Supplementary Data 1); group "Antibiotic" referred to samples treated with antibiotics and other antimicrobial compounds; group "Respiration" referred to hypoxia, reaeration, peroxide stress and nitric oxide stress; group "Genetic manipulation" referred to knockdown, knockout, complementation and over-expression of a gene; group "Nutrient" referred to alterations in carbon sources or other nutrient conditions; group "Infection" referred to samples isolated from ex vivo or in vivo infection models; group "Control" referred to the untreated control

samples of each study. tSNE is archived using the R package 'Rtsne' with the following parameters: dims = 2, PCA = True, max_iter = 100, theta = 0.4, perplexity = 20, verbose = False.

## Estimation of transcriptional plasticity (TP)

MinMax was calculated by subtracting the minimum $\log_2$ (RPKM + 1) from the maximum $\log_2$ (RPKM + 1) for each gene. IQR was calculated by subtracting the 25th percentile of $\log_2$ (RPKM + 1) from the 75th percentile of $\log_2$ (RPKM + 1) for each gene. Considering the underlying association between the variance and the mean of a gene's expressions[31,49,50], the initial standard deviation (SD) measures were calibrated by an estimated global trend between the SD and the mean $\log_2$ (RPKM + 1). This global trend was estimated using a local polynomial regression model (LOESS or Locally Estimated Scatterplot Smoothing) with a large sampling window with the R package stats (version 4.0.2; span = 0.7, degree = 1). A gene's adjusted SD is defined as the sum of this gene's corresponding SD residual of the LOESS fit and the global average of the LOESS fitted SD measures.

## Evaluation of the robustness of expression variation metrics

To evaluate the robustness of the three expression variation metrics, MinMax, IQR, and adjusted SD, we performed a bootstrapping analysis. Specifically, a subset of N (N = 10, 20, 30, 50, 100, 200, 300, 500, or 800) samples were randomly drawn from the dataset, and a Pearson's correlation coefficient (r) was calculated for each metric (MinMax, IQR, or SD) by comparing the randomly sampled output and the corresponding metrics measured using the full dataset. This process was repeated for 100 times for each N, and the means and the standard deviations of the coefficients (r) were depicted in Fig. 1e.

## Enrichment analysis of high-TP genes

To identify high-TP genes, a density curve of adjusted SD was determined with a Gaussian kernel density function using the R package stats (version 4.0.2), and the high-TP subgroup consisted of genes whose TP measures were higher than the upper threshold defined by a probability cutoff of 0.05 based on the probabilistic density estimation of adjusted SD. Gene essentiality and vulnerability indices were referenced from a recently established work that leveraged genome-wide CRISPR interference (CRISPRi) and deep sequencing to render a comprehensive quantification of the effect of differential transcriptional repression on cellular fitness for nearly all Mtb and Msm genes[24]. Enrichment analysis of high-TP genes was performed using the DAVID online server, and enrichment results with FDR (false discovery rate) <0.1 were considered significant.

## Mycobacteria core genome

Homologous genes of mycobacteria including Mtb, Msm and Mab were identified by J.A. Judd et al.[22]. Homologous genes that existed in all three mycobacteria were identified as core genes (Fig. 2c).

## Collection of gene features

**Gene length.** To identify significant gene features that potentially contribute to TP, we first collected genome annotations of Mtb genes from NCBI Genome Database (ASM19595v2). Gene length was identified by the difference between the start position and end position for each gene and then divided by the average length of all genes to calculate the normalized length for each gene.

**Codon usage.** codon usage features, including codon adaptation index (CAI), codon bias index (CBI), frequency of optimal codons (Fop), effective number of codons (Nc), A/T/C/G/GC of silent 3rd codon position (A3s/T3s/C3s/G3s/GC3s), hydrophobicity (Gravy) and aromaticity (Aromo) of a protein were calculated based on gene sequences of Mtb H37Rv (ASM19595v2) by using CodonW (http://codonw.sourceforge.net/).

**Base and amino acid composition.** Based on the reference sequence of a gene, we further identified the percentage of each base type as well as percentages of GC content (GC%) and pyrimidine content (CT%) by calculating the number of each base in a gene divided by the gene length. Similarly, we calculated the percentage of each of the 20 amino acids found in the protein products of the 3891 genes.

**Start and stop codon.** According to the reference genome sequence, we identified the first and the last three bases of coding sequence (CDS) for each gene, referring to the start codon and the stop codon, respectively.

**Direction of replication and transcription.** To study the impact of conflict between replication and transcription on TP, we identified whether DNA replication and RNA transcription were in the same or opposite directions for each gene based on the strand and genome location relative to the *dif* site (2,232,640 bp) of the gene. The site of chromosomal segregation (*dif*) was identified by Cascioferro et al.[51]. To be more specific, genes located on the positive strand and before the *dif* site (clockwise) or genes on the negative strand and after the *dif* site would have the same direction of replication and transcription, and vice versa.

**Transcription factors.** Considering the direct influences of transcription factors (TFs) on transcriptional expression, we collected the data on interactions between TFs and their targets from the *MTB* Network Portal (http://networks.systemsbiology.net/Mtb). The data contained the interaction of 4635 TF-target pairs with evidence of ChIP-seq experiments[52] and transcriptional profiling[53], including 136 TFs and 2111 target genes. TF-target pairs were marked with 1 or −1, representing that TF was an activator or a repressor, respectively. We then counted the number of activators and repressors for each target gene based on the TF–target pairs. The number of target genes for each TF was also counted. In addition, interactions between TFs and their targets identified by ChIP-seq were also selected, including the number of targets located at intergenic and intragenic regions for each TF.

**Selective pressure.** Natural selective pressures (indicated as *dN/dS* ratio) on *Mtb* genes were estimated by GenomegaMap, a phylogeny-free statistical approach performed on 10,209 *Mtb* genomes to estimate substitution parameters[25], including the mean values and 95% CIs (Q2.5 and Q97.5) of d*N*/d*S* ratio, transition: transversion ratio, and substitution rate. The mean probability of an d*N*/d*S* ratio higher than 1 (Pr(d*N*/d*S* > 1)) and the number of sites with Pr(d*N*/d*S*) > 1 for each gene were also included.

**Transcription start sites.** Features associated with a gene's transcription start site (TSS) included upstream TSS subtype (leadered or leaderless), the total number of proximal TSS associated with this gene, maximum/minimum TSS coverage, and the corresponding base at the +1 position of each TSS. TSS annotations were adopted from a previous work by Shell and others[54].

**Promoter sequence composition.** For each gene, we first defined its promoter region by locating the nearest TSS relative to this gene's start codon. We took the sequence from 80 bp upstream to 20 bp downstream of the TSS and from the same strand (+strand), then converted them to numerical descriptors in frequency space. The conversion was performed using the MathFeature webserver[55], whereby binary representations of these promoter sequences were firstly transformed into frequency distribution using Discrete Fourier Transformation (DFT), and the statistical metrics in the frequency domain were measured and used for subsequent analysis.

**Operon.** Operons in *Mtb* were predicted by Roback et al.[56]. We calculated the total number of genes of each operon as well as the position in the operon, which was defined as the order of a gene in its operon. Operon length was defined as the sum of the lengths of all genes in the operon.

**Regulon.** Regulons of *Mtb* were identified by Yoo, R. et al.[57]. Regulons with less than three genes and annotated as "Unknown function", "KO", "Single gene" and "Uncharacterized" were removed in Fig. 5b. To identify whether the TPs of the genes in a regulon were significantly higher or lower than the total TPs of the genes in the genome, we performed gene set enrichment analysis (GSEA) with the R package clusterProfiler (version 3.16.1) to calculate normalized enrichment score (NES) and adjusted the *p*-value for each regulon. NES represents the overall level of TP amplitude of a regulon, whereby higher positive NES values mean higher overall TP and lower negative NES values mean lower overall TP.

**Distance to Ori.** To calculate a gene's distance to the DNA replication initiation site (ori), we first defined its CDS centroid $c_i = (P_{start} + P_{end})/2$, here $P_{start}$ and $P_{stop}$ refer to the genome locus of the beginning and the end of this gene. This gene's distance to *ori* was then described by the following formula:

$$\text{Distance to Ori} = \frac{G}{2} - \left| \frac{G}{2} - c_i \right|$$

where *G* represents the genome size of *Mtb*, which is 4,411,532 bp.

**Vulnerability Index.** Vulnerability Indices for *Mtb* and *Msm* genes were referenced from work by Bosch and others[24].

**Other mycobacteria.** Gene length and GC% of *Msm* and *Mab* were collected from mc²155 and ATCC 19977 genome annotation files derived from Mycobrowser (https://mycobrowser.epfl.ch).

## Machine learning model

To assess the importance of different gene features in determining the TP, we leveraged the recently advanced LightGBM, a decision-tree ensemble model, to perform a multiparametric regression analysis of the 3891 genes and the corresponding 119 features[58]. This was achieved using the Python-compiled *lightgbm* package (version 3.3.2) with the following parameters: *objective* = 'regression', *num_leaves* = 30, *learning_rate* = 0.015, *n_estimators* = 200, *feature_fraction* = 0.75, *max_depth* = 12, *max_bin* = 10, *bagging_fraction* = 0.75, with the remaining parameters set to default. 3891 genes were randomly divided into test and training sets in a ratio of 4:6 using "train_test_split" function from *sklearn*. Then, the LightGBM regression model was trained by training sets with the same parameters mentioned above. To evaluate the performance and robustness of the trained model, the genes were randomly split into test and training groups 100 times, and the importance of each feature and performance ($R^2$) accuracy of the predicted TP with the TP in the test sets were calculated for each time, as shown in Figs. 3c and S4a, respectively.

LightGBM model predicted four robust features, which were operon size, gene length, activating regulator number and GC content. We also performed a support vector machines (SVM) model to assess the predictive power of these 4 features. This was archived using the R package '*e1071*' with the following parameters: *types* = 'eps-regression', kernel = 'radial', degree = 3, cost = 1, gamma = 0.25, coef0 = 0, epsilon = 0.1. Genes missing any feature value were removed so that a total of 2016 genes were included in the analysis. Performance of this SVM model is shown in Fig. 3d. The Shapley additive explanations (SHAP) method was then applied to calculate the contribution of each feature to TP values predicted by the SVM model[59]. We performed SHAP

analysis using the R package 'iBreakDown' (version 2.0.1), and the contribution value of each feature to the predicted TP of each gene was determined. As the contribution value can be positive or negative, representing the portion of the feature making the predicted TP value of a gene higher or lower than the average predicted TP value of all 2016 genes, respectively, the absolute contribution value was taken (Fig. S4b). To test whether there were co-variants among the four features (Fig. 3c) found to affect TP, pairwise Spearman's correlation coefficients were calculated using the R package *stats* (Fig. S4c).

## Differential expression analysis

Differential expression analysis in Fig. S9 was performed on 127 sets of RNA-seq data that contained experimental treatment and corresponding control samples from the total 894 samples by using Linear Modeling for Microarray Data (Limma) package[60]. Genes with adjusted *p*-value < 0.05 were retained for further analysis.

## Statistical analysis

Pearson's correlation coefficients and the corresponding *p* values (Figs. 3d, 6a, b, S2a, b, S2d, S6d, S8c, S9c) were calculated using the R package *stats*; Spearman's correlation coefficients (Figs. 2f, 4a, c, 5d, e, 6f, S1i, S5a, S5e, S6, S7) were calculated using the R package *stats*. The non-parametric unpaired Wilcoxon test was used to make un-paired comparisons and to render the *p* values depicted in Figs. 1e, 2c, d, 4d, 4f, g, 5a, 6c, S5f, g. Paired Wilcoxon test was used in Fig. S1b.

## Reporting summary

Further information on research design is available in the Nature Portfolio Reporting Summary linked to this article.

## Data availability

No primary data has been generated in this study. RNA-Seq data sources are listed in Supplementary Data 1. The conditions of 894 samples are annotated in Supplementary Data 1. The integrated transcriptional profile containing 3891 genes and 894 samples is available in Supplementary Data 2. TP and descriptive statistics for gene expression levels are available in Supplementary Data 3. Collected genetic features are listed in Supplementary Data 4. High-TP genes and their enrichment results are listed in Supplementary Data 5. Regulon genes of *Mtb* are listed in Supplementary Data 6. TP data of *Msm* and *Mab* are available in Supplementary Data 7. Benchmark of DEGs based on TP data of *Mtb* are shown in Supplementary Data 8. Source data are provided with this paper (https://github.com/ChengBEI-FDU/Transcriptional_Plasticity/tree/main/source_data).

## Code availability

Code for data analysis in this study is available from the following GitHub repository, https://github.com/ChengBEI-FDU/Transcriptional_Plasticity (https://doi.org/10.5281/zenodo.10846626).

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

## Acknowledgements

We thank Dr. Howard E. Takiff for editing the manuscript. This work was supported by the National Institutes of Health (grant P01 AI132130 to S.M.F.). This study was also financially supported by the National Natural Science Foundation of China (82272376 to Q.G.) and Shanghai Municipal Science and Technology Major Project (ZD2021CY001to Q.G.).

## Author contributions

C.B., J.Z. and Q.L. designed the research. Q.G. and Q.L. supervised the research. C.B., M.G. and J.Z. analyzed the data. C.B., J.Z. and Q.L. interpreted the results. P.H.C., E.J.R. and S.M.F. were involved in discussions of the results. C.B., J.Z. and Q.L. wrote the paper, which was edited by P.H.C., E.J.R., S.M.F. and Q.G.

## Competing interests

The authors declare no competing interests.
