## [Peer Review File · Nature Communications]

REVIEWER COMMENTS

Reviewer #1 (Remarks to the Author):

This study presents a meta-analysis of *Mycobacterium tuberculosis* transcriptome data that focuses on characterizing a property that the authors call the 'transcriptional plasticity' of *M. tuberculosis* genes. Defined as the mean-adjusted standard deviation of each gene's expression level across conditions, the authors find that the magnitude of this transcriptional plasticity value is anti-correlated with Mtb gene essentiality or vulnerability (essential and vulnerable genes tend to have less transcriptional plasticity than genes that are non-essential or even growth-advantageous when disrupted). The authors also find associations between transcriptional plasticity and genomic and regulatory features, including: gene length, the number of genes within an operon, GC content, and number of known regulators. The authors also find that the transcriptional plasticity of a gene is a property that is conserved across mycobacterial species (when compared with *M. smegmatis* and *M. abscessus*). This descriptive study presents an interesting property of gene expression; however, the manuscript would benefit from a clearer description of the broader significance of transcriptional plasticity, as well as a more detailed assessment of batch effects and TMM-based batch effect correction.

Major concerns:

- It is unclear how the property of transcriptional plasticity for a particular gene should be used to inform future research investigations. Although the authors found statistical associations between transcriptional plasticity and genomic and regulatory features, it is not clear whether any of those associations lead to any predictive power of underlying biology. For example, how would the knowledge of the extent of transcriptional plasticity of a gene of unknown function inform on that gene's function or inform on designing follow-up experiments to study that gene's role in *M. tuberculosis* physiology? Or alternatively, if there were two genes of unknown function with different transcriptional plasticities (e.g. if one of the genes were in the top 5% percentile of transcriptional plasticity and the other was not), under what circumstances would the gene with high transcriptional plasticity be prioritized for further study, and what circumstances would the gene with lower transcriptional plasticity be prioritized for further study? Additionally, if future authors were to calculate transcriptional plasticity for some other organism, what would similarities or differences in transcriptional plasticity between homologous genes indicate from a biological standpoint? Some additional clarification in the text about how to interpret the transcriptional plasticity metric would be helpful for understanding the impact of this measurement.

- The transcriptional plasticity metric could potentially be severely impacted by batch effect issues in the transcriptome compendium analyzed. Although TMM was used to help address batch effects, it is not clear the extent to which this processing mitigated the issue. Including a visualization of the data (e.g. PCA) with the different batch labels identified both pre and post TMM normalization would help to clarify how much TMM has contributed to mitigating batch effects, and performing the sub-sampling and analysis in a batch-specific way could help to clarify how much batch-to-batch variability is contributing to the calculated value of transcriptional plasticity.

Reviewer #2 (Remarks to the Author):

In this manuscript, the authors have investigated the transcriptional plasticity (TP) of *Mycobacterium tuberculosis* genes using previously published RNA sequencing data derived from 73 different conditions, totaling 894 datasets. They calculated TP for every gene, excluding genes shorter than 150 base pairs. The authors also explored the correlation of TP with gene function, length, GC content, essentiality, vulnerability, and location (whether they are encoded on monocistronic or polycistronic mRNA). Additionally, they generated a machine learning model to predict TP for each gene and tested 57 different gene features regarding their importance in the prediction process. Moreover, they extended a part of the investigation on other *Mycobacterium* species, *M. smegmatis* and *M. abscessus*. They concluded that TP is correlated with gene function, length, location, GC content, and GC content deviation, essentiality, vulnerability.

Overall, the manuscript offers a comprehensive investigation into TP in *M. tuberculosis*, utilizing a rich dataset and advanced analytical methods. The findings shed light on the nuanced relationship between gene characteristics and transcriptional plasticity, contributing to our understanding of this essential biological process. I would like to express my appreciation for the well-written main text. However, I encountered some challenges when attempting to interpret both the main figures and the supplementary figures. This difficulty arose from the inconsistency in the use of terms across several plots, where different terms were employed to describe the same. Additionally, in some instances, they refer to wrong figure or the figure legends were either absent or lacked sufficient information, which further compounded the issue. Moreover, the abstract seems to be somewhat incomplete and doesn't include several important findings from the main text. For instance, it doesn't mention the SVM model or the analysis of TP in genes encoded on monocistronic or polycistronic mRNAs, both of which are significant aspects of the study.

While I largely agree with most of the conclusions drawn from the analysis, I do have some reservations regarding the conclusion concerning the correlation between TP and gene function. In my view, this conclusion appears to lack sufficient supporting evidence. Additionally, in certain instances, contradictory results were obtained regarding high and low TP values for specific biological functions, as I have tried to explain in the 'Major comments' section of my review. Furthermore, I found the findings regarding TP and gene location to be particularly intriguing, and I believe it deserves greater emphasis within the main text. There is potential for further exploration through additional analyses, as outlined in my major comments.

Major comments

- In line 110, the authors refer to Fig. S2 when mentioning about the biological functions and high TP. However, Fig. S2 shows the expression (Z-score) of the 195 high-TP genes. This being said, I do not understand what kind of conclusion should be drawn from this Fig. S2. Does this heatmap show any significant pattern associated to any functional group? If yes, this should be clearly shown on the figure. If not, I would rather remove it.
- In line 95, the authors state that there is a high degree of correlation between MinMax, IQR, and adj-SD. Looking at the Fig. S1c, I do not see a 'high degree' of correlation between IQR and MinMax and Fig. 1d also support this. Therefore, this statement should be corrected to 'different degree of' correlation or

clearly describe the differences in different comparisons.

- In Fig. S1d and e, the X-axis of the plots are presented as 'count' and 'density', respectively. Do they mean the same thing? If yes, the same should be used in both panels. If not, the authors should describe the count and density terms and why they used different in those two panels.
- In line 110-112 and Fig. 2a, the authors concluded that the 195 high TP genes were enriched in certain biological functions according to DAVID database. How many of the 195 genes were assigned to at least one of those biological functions? The authors should refrain to draw such conclusion without knowing what percentage of the 195 genes are presented in this analysis. Without knowing this, I cannot evaluate the output of the analysis. Therefore, they should provide a gene list with their assigned biological function. It could be somewhat similar to Fig S7d but with gene lists in each category. Moreover, in Fig 2a the authors conclude that the high TP genes were enriched in cell wall and hypoxia while in Fig 2b the lowest TP genes were enriched in respiration and cell wall and cell wall processes. These two findings are quite contradictory and raise the question 'Do the genes involved in cell wall and respiration/hypoxia primarily associated with highest TP genes or lowest TP genes? Perhaps it is difficult to associate TP with biological functions and the conclusion drawn from such analysis lacks solid evidence. Moreover, most of the functional categories in Fig. 2b seem to be associated with low TP, which might be simply due to the fact that most of Mtb genes have low TP. See also my comments to Fig S7d below.
- In line 154-155, the authors consider that $r < 0.21$ shows no degree of correlation. However, in many of the correlation analyses in both main figures and supplementary figures (Fig S5, S6, Fig. 6a, b, and f) the authors accept a degree of correlation for even lower r values. This is confusing and contradictory when it comes to interpreting the output of an analysis. I think $r = 0.21$ is still a degree of correlation. For example; the authors think $r = 0.22$ is an apparent correlation for TP and GC% deviation in Fig S4a. Therefore, the sentence in lines 154-155 should be corrected.
- Without being sure, is it possible to take the base substitution rate of regulatory regions of genes and compare this to TP? This is because the plasticity of gene expression is assumed to be primarily linked to the regulatory region of the genes.
- In prokaryotes, the DNA concentration near the replication origin is typically higher than that near the chromosome terminus due to the bidirectional replication process in replicating bacteria. Considering that the dataset used in this study comprises samples from 73 different conditions, presumably including samples from both the exponential and stationary phases, I am curious whether the genomic location, defined as the distance from the replication origin, could be a valuable feature for predicting TP in the SVM model. It might be worthwhile to incorporate this feature into the current model to assess whether its inclusion enhances TP prediction. Furthermore, I also wonder whether gene vulnerability could serve as another useful feature in the prediction model. It might be worth exploring the potential inclusion of gene vulnerability as a feature to improve the accuracy of TP prediction.
- The analysis performed on TP of genes belonging to monocistronic and polycistronic mRNAs is very much striking. Therefore, I think Fig. S4d should be lifted to main text. I wonder if the authors noticed that the correlation between TP of genes in the same operon gets lower when the distance between genes increases. For example, in Fig. S4d (if I interpreted the figure correctly), the R for the first gene in the operon; 1st to 2nd = 0.56, 1st to 3rd = 0.45, and 1st to 4th = 0.41 and the second gene in the operon; 2nd to 3rd = 0.63, 2nd to 4th = 0.44. As you can see with those examples as the distance increases the correlation gets lowered. If this is correct, why is it so?
Is the location of the genes in operons related to their TP? Could authors test this further? One could check the TPs of genes in different locations in the operon to answer this question.

- TP to GC% deviation for whiB4 and sigH, TP to gene length for whiB1, zur, sigD and Rv1828 were repeated in Fig. 5d and e and in Fig. S5 and Fig. S6. Moreover, while the R values are the same in two places, the p-value changes. Why is this difference in same analysis shown in two different figures?
- In Fig. S7d, the conclusion made on the functional enrichment analysis is based on the total of 23 genes (if there is no common genes in the two groups) out of 316 genes for Msm and 51 genes (if there is no common genes in the two groups) out of 242 genes for Mab. I think the conclusion made on the function and TP has no strong evidence. Therefore, I do not see association of TP to any biological function.
- I would like to express my appreciation for the authors' suggestion to incorporate TP into the analysis of differential expression, along with the provision of data in Table S6. However, I believe it would greatly benefit the readers if Table S6 were accompanied by a more detailed description. In lines 298-300, the authors mentioned the 5th and 95th percentiles without specifying the percentile of what exactly, which might require clarification for better comprehension.

Minor comments:

- What does the X-axis of hspX, rpoB, and lpqM plots in In Fig 1c present?
- The figure legend for Fig. S1b is missing. This should be added. The same legend there is '@t' at the end of sentence describing d panel. This should be corrected.
- In Fig. S1c, aSD was used instead of adj-SD. To avoid confusion, it should be adj-SD as in Fig1d instead of aSD.
- In line 107, the statement about the sample size reduction to 10 'genes', I guess is wrong. Should it be 'samples' or 'conditions' instead of 'genes'?
- In line 115, the number of core genes and variable genes should be mentioned and it would be more appropriate to place the reference here (which is now in Materials and Methods).
- The figure legend for Fig. 2b states that 'The X-axis is presented on a log scale'. Is it correct for Fig. 2b or should it be for Fig. 2a?
- Pearson and Spearman correction coefficient are represented randomly different (R, R2, r) in the figures and text. It should a consistent representation in overall figures and text.
- In Fig S5 and S6 legends, it should be 'Spearman's correlation coefficient' not 'correction'
- In Fig 4c and Fig. S4a the X-axis of the plots show SD of GC% and GC% deviation, respectively. Do these two terms refer to the same? If yes, one only should be used to avoid confusion. If not, they should describe what they mean
- I did not quite understand why the authors wanted to pick 50 quantiles and SD GC% on ranked bins in Fig. 4c. They should explain.
- The X-axis in Fig. S5 should be gene length.
- In Fig. S7a, the number 20 should 200 in X-axis.
- In line 221, they should refer to Fig. 2a instead of Fig. 3a
- In some cases, gene names were written as protein names such as in Fig. 5d and e. For example; it should be sigD instead of SigD.

Reviewer #3 (Remarks to the Author):

This paper describes a meta-analysis of hundreds of Mycobacterium tuberculosis RNA-seq datasets to

explore the genetic basis of transcriptional plasticity. Overall I like the approach taken, and the manuscript is well laid out. However I have concerns about underlying data quality / pre-processing that may impact conclusions.

Major concerns:

It seems possible that the observed relationships between operon length / gene length / GC and Transcriptional plasticity are due to technical artifacts in the underlying RNA-seq datasets (primarily differences in coverage uniformity or GC bias). These biases would also effect data from different species giving the appearance of conserved patterns of TP (figure 6). Extra analyses are required to rule out this possibility.

-Does the data normalization approach used sufficiently account for all batch effects? It likely works well for comparison of the same genes across different samples, but I am concerned that there may be biases remaining that undermine comparisons across different genes (e.g. genes of different length or GC content)

-How reliable is the correlation between TP and operon length and gene length? I wonder if this could be an artifact of poor coverage uniformity in some underlying datasets (e.g. where RNA degradation results in failure to cover 3' end of long genes or operons in some datasets). You could check RNA-seq coverage uniformity of datasets (eg by visualization in IGV, or by making metagene using deepTools), then segregate the data based on gene body coverage uniformity metrics. Compare the gene length-TP relationship in conditions exhibiting the most uniform coverage against those with the least uniform coverage.

-Similarly, how reliable is the correlation with TP and GC content? Regions of GC are notoriously problematic for library prep and sequencing and are often associated with coverage biases. Again try to assess the GC coverage bias of individual samples and check if TP-GC relationship is robust.

I like the overall approach used to explore the genetic basis of TP (figure 3), but R2 of 0.17 seems disappointingly low, and most of the variance in TP remains unexplained. Could the table of features be expanded (e.g. to include specific TFBS annotations / promoter sequence composition / other?). Investigation of some of the regions in figure 1.e. with very high TP could provide additional clues.

Minor comments / concerns:

-Introduction, line 24-26. The concept and definition of Transcriptional Plasticity was not clear to me and took several readings. It would be helpful to give a clearer more expansive explanation (e.g. TP can be visualized as the extent to which gene expression levels shift across multiple conditions. For instance, a gene with high TP would exhibit significant changes in its expression levels under different environmental stresses, while a gene with low TP would show relatively stable expression regardless of the environment.)

-Results, line 89-92. Can you speculate on why hspX and ipqM might have different TP?

-Fig 1.e. It would be interesting to know what genes are at genomic regions with spikes in extreme TP

e.g. at 2, 2.25, 3, 3.5MB. Are these operons? Are they horizontally transferred sequences? Visual inspection of these regions may provide clues about causes that are not explained by the SVM model.

-Line 105: "Using a bootstrap approach similar to that described in Fig.1d, we found that the 195 high-TP genes in the top 5% percentile demonstrated consistently high TP even when the sample size was reduced to just 10 genes (Fig. S1d, e)". Should this sentence be "reduced to just 10 samples"?

-Line 127: 'We hypothesized that high-TP genes may promote phenotypic diversification that confers a selective advantage in the ability of Mtb to survive in fluctuating environments, and therefore these genes might accumulate mutations more rapidly than the rest of the genome.' - I don't follow this. Could it not be argued that we expect these genes to be more conserved?

Point-by-point responses to reviewers' comments

We appreciate the reviewers' interest in this work and their comments for helping improve this manuscript. In this revision, we performed additional analyses to address those comments or suggestions. In brief, these analyses include:

- 1) Addressing the potential influence of batch effects, GC% and read coverage uniformity on TP estimation.
- 2) Demonstrating the effectiveness of normalizing TP in differential gene expression analysis.
- 3) Implementing new genetic features (such as gene's distance to Ori and gene vulnerability) in TP prediction model.
- 4) Exploring the underlying association between TP and biological functions.

We believe that these analyses have further strengthened our major findings. Below, we have provided point-by-point responses to the reviewers' comments. Our responses are shown in "blue", the quoted text from the manuscript is shown in "grey", and the changed text in the manuscript is shown in "red".

Reviewer #1 (Remarks to the Author):

This study presents a meta-analysis of Mycobacterium tuberculosis transcriptome data that focuses on characterizing a property that the authors call the 'transcriptional plasticity' of M. tuberculosis genes. Defined as the mean-adjusted standard deviation of each gene's expression level across conditions, the authors find that the magnitude of this transcriptional plasticity value is anti-correlated with Mtb gene essentiality or vulnerability (essential and vulnerable genes tend to have less transcriptional plasticity than genes that are non-essential or even growth-advantageous when disrupted). The authors also find associations between transcriptional plasticity and genomic and regulatory features, including: gene length, the number of genes within an operon, GC content, and number of known regulators. The authors also find that the transcriptional plasticity of a gene is a property that is conserved across mycobacterial species (when compared with M. smegmatis and M. abscessus). This descriptive study presents an interesting property of gene expression; however, the manuscript would benefit from a clearer description of the broader significance of transcriptional plasticity, as well as a more detailed assessment of batch effects and TMM-based batch effect correction.

Reply: We thank the reviewer for these valuable comments. Indeed, batch normalization has been a major challenge for studies that integrate RNA-Seq data from different studies. As suggested by the reviewer, we performed additional analyses in individual batches and found that the TPs estimated from individual batches exhibited strong positive correlations with those estimated from the entire dataset. Please see the detailed analyses and results in the following responses.

Major concerns:

- It is unclear how the property of transcriptional plasticity for a particular gene should be used to inform future research investigations. Although the authors found statistical associations between transcriptional plasticity and genomic and regulatory features, it is not clear whether any of those associations lead to any predictive power of underlying biology. For example, how would the knowledge of the extent of transcriptional plasticity of a gene of unknown function inform on that gene's function or inform on designing follow-up experiments to study that gene's role in M. tuberculosis physiology? Or alternatively, if there were two genes of unknown function with different transcriptional plasticities (e.g. if one of the genes were in the top 5% percentile of transcriptional plasticity and the other was not), under what circumstances would the gene with high transcriptional plasticity be prioritized for further study, and what circumstances would the gene with lower transcriptional plasticity be prioritized for further study? Additionally, if future authors were to calculate transcriptional plasticity for some other organism, what would similarities or differences in transcriptional plasticity between homologous genes indicate from a biological standpoint? Some additional clarification in the text about how to interpret the transcriptional plasticity metric would be helpful for understanding the impact of this measurement.

Reply: The reviewer asked about the significance and future applications of TP, and we agree that providing additional justifications and clarifications would improve this work. Below, we have addressed each of the reviewer's comments separately.

(1) The significance of TP in future research investigation.

We believe that the most applicable significance of TP, at this stage, is its potential use as a benchmark for identifying differentially expressed genes. In standard RNA-Seq studies, genes that exhibit large expression changes (e.g. >2 fold) tend to receive more attention than ones whose expression changes are less pronounced (e.g. <2 fold). For this reason, high-TP genes are more likely to be prioritized while low-TP genes could be neglected due to their relatively smaller changes in expression. Instead of using a binary fold-change cutoff, we propose incorporating TP into RNA-Seq analysis as a 'normalization' factor for ranking gene candidates. Below, we provide some pilot examples showing how TP may be used for this purpose:

- To normalize TP's effect on fold-change values (**Fig. S9a**), we propose to divide log₂ fold changes (logFC) of a gene by its TP value. By doing this, logFC values of low-TP genes would be divided by smaller numbers, which would result in an increase of low-TP genes' ranking in differentially expressed genes. Indeed, we found the positive correlation between "TP-adjusted logFC" and "TP" was diminished after this normalization (**Fig. S9b**), while the values of logFC and TP-adjusted logFC were still highly correlated (Pearson's correlation coefficient: 0.93) (**Fig. S9c**).
- We further selected one dataset (PRJNA733783) for detailed analysis. This dataset was generated in a single experiment by cholesterol treatment (Pawelczyk et al., Sci Rep., 2021). As expected, logFC values were positively correlated with TP (**Fig. R1a**) and was diminished by TP normalization (**Fig. R1b**). We found that, after TP normalization, "steroid degradation" and "cholesterol metabolism processes" became top-ranking with the most significant FDR values (**Fig. R1c-d**), which could reflect the treatment condition (cholesterol). Besides, more genes involved in cholesterol metabolism were identified as differentially expressed after TP adjustment, including *fadD8*, *fadE28*, *cyp51*, *cyp142*, *hsaC* and *hsaD* (**Fig. R1e-f**), all of which, however, were masked before the TP adjustment.

Taken together, these pilot analyses demonstrate that incorporating TP can diminish the association between TP and expression fold changes, which could help identify meaningful gene candidates. In this revised manuscript, we have added a paragraph in the discussion (lines 371-378). Besides, we are planning a follow-up study to thoroughly assess how TP can be used in future RNA-Seq analysis.

Lines 371-389:

"Alternatively, we propose incorporating TP into RNA-Seq analysis as a normalization factor. In our curated dataset of 894 RNA-Seq samples, we observed that the fold changes of differentially expressed genes (DEGs) were positively correlated with their TP values (Fig. S9a, see Materials and Methods). A potential way to diminish this effect is to divide log₂ fold change (logFC) by the gene's TP. By doing this, logFC values of low-TP genes would be divided by smaller numbers than high-TP genes, which would result in an increase of low-TP genes' ranking in DEGs. After this normalization, the positive correlation between "TP-adjusted logFC" and TP was reduced (Fig. S9b), while the values of logFC and TP-adjusted logFC were still highly correlated (Fig. S9c, Pearson's correlation coefficient: 0.93)."

Fig. S9 (a-b) Correlation between TP and absolute log₂ FC values (a) and absolute TP-adjusted log₂ FC values (b) in 127 differential expression analyses (see *Materials and Methods*). Genes are binned to 10 bins with equal size according to their TPs. Box plots represent median $\pm 1.5 \times$ IQR. **(c)** Correlation between absolute log₂ FC and TP-adjusted log₂ FC.

Fig. R1 (a-b) Correlation between TP and absolute log₂ FC values (a) and absolute TP-adjusted log₂ FC values (b) in a single experiment that refers to cholesterol treatment (PRJNA733783). **(c-d)** Enrichment analyses of up-regulated DEGs identified by original log₂ FC (c) and TP-adjusted log₂ FC (d), respectively. The threshold of log₂ FC for identifying DEGs is 2. **(e)** Correlation between absolute log₂ FC and TP-adjusted log₂ FC in the cholesterol-treatment experiment. Green dots represent newly identified up-regulated DEGs using TP-adjusted log₂ FC. **(f)** 6 up-regulated genes related to cholesterol metabolism are newly identified.

(2) Predicting power of TP in underlying biology? *How would the knowledge of the extent of transcriptional plasticity of a gene of unknown function inform on that gene's function or inform on designing follow-up experiments to study that gene's role in M. tuberculosis physiology?*

We thank the reviewer for raising these questions. Although we have strived to establish a connection between a gene's TP and its biological function, we believe that such relationship might not be explained by one or a few simple rules. Nevertheless, we propose a couple guiding principles in leveraging TP ranks to inform biology:

- Low-TP genes, especially those with moderate to high steady-state expression levels, are particularly interesting as both stringent control of gene expression and the high-level production of RNA and proteins could be resource-consuming, suggesting that the high expression of these genes must confer some fitness advantage to the bacteria. While many of these high-expression (median log₂-RPKM > 7), low-TP (TP < 0.5) genes are indeed essential, we identified a couple of outliers that are annotated as non-essential for *Mtb*. One such example is *Rv0012* (TP=0.47), which encodes a membrane protein that is highly conserved among mycobacterial species but is dispensable for *Mtb*'s growth both *in vitro* and during infection (Sassetti et al., P.N.A.S., 2003; Bosch et al., Cell 2021). However, a recent chemical-genetic screening revealed that transcriptional repression of *Rv0012* substantially sensitized *Mtb* to multiple antibiotics, especially antibiotics that target cell wall biosynthesis, such as Vancomycin (Li, et. al., Nat. Microbiol. 2022). While the biological functions of *Rv0012* remain largely unknown, this example showcases how TP could supplement gene essentiality and transcriptional vulnerability to help guide gene prioritization.
- The expression profiles of low-TP genes are not always invariable, they can also be infrequently variable. One example is the PPE protein encoded by *Rv0265c* which is thought to play an important role in heme uptake. *Rv0265c* had a low TP (TP = 0.59, in the lowest 10% percentile of all genes) but was unexpectedly

highly expressed in the late dormant phase (PRJNA276810, Ignatov et al., BMC Genomics., 2015). Therefore, changes in the expression of low TP genes under specific conditions may provide useful clues about their physiological functions.

In the revision, we have added the following to explain the potential of TP to informing a gene's function in lines 325-336:

"As TP provides an empirical reference for gene expression variability across different conditions, we reason that knowing a gene's TP could have value by suggesting its role in Mtb physiology. For example, while essential genes are generally associated with low TPs, there are some non-essential/invulnerable genes that also exhibit very low TPs. One intriguing example is Rv0012, which is highly expressed (median log₂-RPKM > 7) but has a very low TP (0.47). Rv0012 encodes a membrane protein that is conserved among mycobacterial species but is not essential for Mtb's growth either in vitro or during infection^{24,43}. However, a recent chemical-genetic screening revealed that transcriptional repression of Rv0012 substantially sensitized Mtb to multiple antibiotics, especially antibiotics that target cell wall biosynthesis, such as vancomycin⁴⁴. While it remains unclear why the Mtb genome harbors a group of non-essential, low-TP genes, this example suggests that this unique gene subset warrants closer examination and demonstrates that the TP may be useful as a supplement to gene essentiality and vulnerability for quiding gene candidate prioritization."

(3) Which genes should we prioritize, low-TP or high-TP genes?

Differential expression analysis usually identifies tens or even hundreds of genes whose expression shows significant changes. Prioritizing gene candidates has always been a challenging task, but we believe that the genes with low TPs are worth more attention in future differential expression analyses. Low-TP genes' expression are either very stable or vary only under few conditions. However, if the expression of low-TP genes exhibits significant changes when subject to a particular growth condition or stress, this could be a useful clue about its functional role.

Additionally, we should be cautious about differentially expressed high-TP genes, because their frequent variations in expression are sometimes seen even in the "control" samples without treatment. The differential expression of these genes (e.g., DosR regulon) may be caused by subtle perturbations, such as in the oxygen concentration, that are hard to precisely control from one experiment to another. Normally, one can recognize these experimental artifacts by checking the consistency of a gene's expression in both treatment and control samples.

In this revised manuscript, we have added these discussions in lines 338-347.

"In future differential expression analyses, significant changes in the expression of genes with low TPs should be given more attention, as deviations from their usual level of expression could implicate them in the bacteria's response to particular experimental settings. In contrast, caution should be exercised regarding the differential expression of high-TP genes, as they often exhibit changes in expression even in "control" samples subjected to "no treatment". For example, changes in the expression levels of genes in the DosR regulon are frequently observed in control samples, presumably due to subtle differences from one experiment to another in parameters such as oxygen concentration. Therefore, it is worthwhile checking the validity of high-TP values by looking at the consistency of expression amongst samples within a group, such as either treated or untreated."

(4) Additionally, if future authors were to calculate transcriptional plasticity for some other organism, what would similarities or differences in transcriptional plasticity between homologous genes indicate from a biological standpoint?

The genus *Mycobacteria* contains different species that have adapted to distinct ecological niches. We posit that the homologous genes, even those with the same function in different species, could exhibit disparate TP levels across *Mycobacteria* species corresponding to their niche adaptation. In support of this scenario, we found that whereas most homologues exhibit similar TPs in *Mtb*, *Msm* and *Mab*, there is a small proportion whose TP levels vary in the different species (**Fig 6a-b**, **Fig S8c**). Among the top 5% *Mtb* genes with the most

different TPs compared to the other two mycobacteria species, we noticed that genes involved in amino acid biosynthesis had significantly higher TPs in *Mtb* than in *Msm* and *Mab* (Fig. S8f), including biosynthesis of arginine (*argD*, *argJ*), chorismate (*aroF*, *aroK*), and leucine (*leuC*, *leuD*) (Fig. S8g). Intriguingly, amino acid biosynthesis, such as arginine synthesis, was found to be involved in the responses of *Mtb* to oxidative stress, DNA damage and host immune pressure (Tiwari et al., *Proc Natl Acad Sci U S A.*, 2018). Therefore, the increased TP levels of these genes in *Mtb* could reflect an adaptation to the within-host microenvironment.

In the revised manuscript, we have added a paragraph in the main text to address this point (lines 255-261):

"On the contrary, although the TP landscape overall demonstrated conservation across the three species, the outliers—genes exhibiting distinct TPs in different Mycobacterium species—could be related to their niche adaptation. For instance, compared to Msm and Mab, we observed significantly higher TPs in genes related to amino acid biosynthesis in Mtb (Fig. S8f-g). Intriguingly, amino acid biosynthesis, such as arginine synthesis, has been found to be involved in Mtb's responses to oxidative stress, DNA damage, and host immune pressure³¹. Therefore, the elevated TP levels of these genes may represent an adaptation to the host microenvironment."

Fig. S8 (f) Enrichment analysis of top 5% *Mtb* genes that showed higher (top) or lower (bottom) TPs than the homologous genes in *Msm* and *Mab*. **(g)** TP of amino-acid synthesis genes is higher in *Mtb* (red) than in *Msm* (orange) and *Mab* (blue).

- The transcriptional plasticity metric could potentially be severely impacted by batch effect issues in the transcriptome compendium analyzed. Although TMM was used to help address batch effects, it is not clear the extent to which this processing mitigated the issue. Including a visualization of the data (e.g., PCA) with the different batch labels identified both pre and post TMM normalization would help to clarify how much TMM has contributed to mitigating batch effects, and performing the sub-sampling and analysis in a batch-specific way could help to clarify how much batch-to-batch variability is contributing to the calculated value of transcriptional plasticity.

Reply: We appreciate these suggestions. In this revised manuscript, we have added figures in Fig. S1 to illustrate the PCA before and after TMM normalization across the entire datasets. As expected, the PCA distribution after TMM normalization is less dispersed than before TMM normalization, suggesting that TMM normalization mitigated the batch effects. We also added a few panels to demonstrate the global gene expressions across samples before and after TMM normalization (Fig. S1a-f), which again indicate that TMM normalization reduced the batch effects.

We also followed the reviewer's suggestion by performing sub-sampling analysis using single batches that had large numbers of RNA-Seq samples (PMID 32348771, 29511081 and 32916109). We calculated TPs in each individual subset and compared these to the TPs estimated from the entire dataset (Fig. S2c). We found that the TPs estimated from subsets showed positive correlations with TPs calculated from all 894 samples (Fig. S2d). Moreover, we found that the TPs estimated from bootstrap analysis were also positively correlated with TPs calculated from the entire dataset (Fig. 1e). Therefore, the subset analyses provide confidence that our TP estimation is robust to potential batch effects associated with different experiments.

In the text of the revised manuscript, we have added a few sentences to address batch effects (Lines 119-129):

"Because other technical factors could potentially affect the estimation of TPs, such as read coverage uniformity and GC content-associated sequencing bias¹⁸, we conducted constrained analyses to assess their influence by controlling for each factor (Fig. S2a-b). TPs calculated from subsets of samples grouped by their

degrees of mRNA coverage uniformity or GC% preference were still highly correlated with TPs calculated from the entire dataset (Fig. S2a-b). Additionally, to exclude the possibility that our results were biased by technical factors associated with experimental batches, we applied our analyses to three independent BioProjects with relatively large numbers of samples, and estimated TPs within each of these batches (Fig. S2c). We found that TPs calculated from individual batches, despite their smaller sample sizes, still showed a high correlation with the TPs calculated from the entire dataset (Fig. S2d). Together, these analyses give us confidence that the TP estimation is robust to technical biases associated different experimental batches.”

Fig. S2 (c) The samples from three independent studies were selected to measure batch-specific TP. **(d)** The TP calculated with the samples from each individual study (subset-TP) is significantly correlated with original TP based on the whole dataset.

Reviewer #2 (Remarks to the Author):

In this manuscript, the authors have investigated the transcriptional plasticity (TP) of *Mycobacterium tuberculosis* genes using previously published RNA sequencing data derived from 73 different conditions, totaling 894 datasets. They calculated TP for every gene, excluding genes shorter than 150 base pairs. The authors also explored the correlation of TP with gene function, length, GC content, essentiality, vulnerability, and location (whether they are encoded on monocistronic or polycistronic mRNA). Additionally, they generated a machine learning model to predict TP for each gene and tested 57 different gene features regarding their importance in the prediction process. Moreover, they extended a part of the investigation on other *Mycobacterium* species, *M. smegmatis* and *M. abscessus*.

They concluded that TP is correlated with gene function, length, location, GC content, and GC content deviation, essentiality, vulnerability.

Overall, the manuscript offers a comprehensive investigation into TP in *M. tuberculosis*, utilizing a rich dataset and advanced analytical methods. The findings shed light on the nuanced relationship between gene characteristics and transcriptional plasticity, contributing to our understanding of this essential biological process. I would like to express my appreciation for the well-written main text. However, I encountered some challenges when attempting to interpret both the main figures and the supplementary figures. This difficulty arose from the inconsistency in the use of terms across several plots, where different terms were employed to describe the same. Additionally, in some instances, they refer to wrong figure or the figure legends were either absent or lacked sufficient information, which further compounded the issue. Moreover, the abstract seems to be somewhat incomplete and doesn't include several important findings from the main text. For instance, it doesn't mention the SVM model or the analysis of TP in genes encoded on monocistronic or polycistronic mRNAs, both of which are significant aspects of the study.

While I largely agree with most of the conclusions drawn from the analysis, I do have some reservations regarding the conclusion concerning the correlation between TP and gene function. In my view, this conclusion appears to lack sufficient supporting evidence. Additionally, in certain instances, contradictory results were obtained regarding high and low TP values for specific biological functions, as I have tried to explain in the 'Major comments' section of my review. Furthermore, I found the findings regarding TP and gene location to be particularly intriguing, and I believe it deserves greater emphasis within the main text. There is potential for further exploration through additional analyses, as outlined in my major comments.

Major comments

- In line 110, the authors refer to Fig. S2 when mentioning about the biological functions and high TP. However, Fig. S2 shows the expression (Z-score) of the 195 high-TP genes. This being said, I do not understand what kind of conclusion should be drawn from this Fig. S2. Does this heatmap show any significant pattern associated to any functional group? If yes, this should be clearly shown on the figure. If not, I would rather remove it.

Reply: We thank the reviewer for pointing out this confusing part. Our intention was to show the clustering patterns of high-TP genes, however, the previous version of Fig. S2 didn't show the enrichment of functional categories of those 195 high-TP genes. In the updated figure (now Fig. S3), we added gene names and enriched functional categories to the heatmap.

- In line 95, the authors state that there is a high degree of correlation between MinMax, IQR, and adj-SD. Looking at the Fig. S1c, I do not see a 'high degree' of correlation between IQR and MinMax and Fig. 1d also support this. Therefore, this statement should be corrected to 'different degree of' correlation or clearly describe the differences in different comparisons.

Reply: Following the reviewer's suggestion, we modified the text from *"we found a high degree of correlation between MinMax, IQR, and adj-SD"* to *"we found significant correlations between MinMax, IQR, and adj-SD with different levels of correlation coefficients"* in lines 111-112.

- In Fig. S1d and e, the X-axis of the plots are presented as 'count' and 'density', respectively. Do they mean the same thing? If yes, the same should be used in both panels. If not, the authors should describe the count and density terms and why they used different in those two panels.

Reply: Yes, both terms represent the number of genes within a given TP range. In this revised manuscript, we have changed this figure (now Fig. S2e) to a density plot. We appreciate this suggestion.

- In line 110-112 and Fig. 2a, the authors concluded that the 195 high TP genes were enriched in certain biological functions according to DAVID database. How many of the 195 genes were assigned to at least one of those biological functions? The authors should refrain to draw such conclusion without knowing what percentage of the 195 genes are presented in this analysis. Without knowing this, I cannot evaluate the output of the analysis. Therefore, they should provide a gene list with their assigned biological function. It could be somewhat similar to Fig S7d but with gene lists in each category. Moreover, in Fig 2a the authors conclude that the high TP genes were enriched in cell wall and hypoxia while in Fig 2b the lowest TP genes were enriched in respiration and cell wall and cell wall processes. These two findings are quite contradictory and raise the question 'Do the genes involved in cell wall and respiration/hypoxia primarily associated with highest TP genes or lowest TP genes? Perhaps it is difficult to associate TP with biological functions and the conclusion drawn from such analysis lacks solid evidence. Moreover, most of the functional categories in Fig. 2b seem to be associated with low TP, which might be simply due to the fact that most of Mtb genes have low TP. See also my comments to Fig S7d below.

Reply: We thank the reviewer for these suggestions. Among the 195 high-TP genes, 55 genes were assigned to at least one biological function in Fig. 2a. Among them, 15 were assigned to "Response to hypoxia", 11 to "Host immune response", 6 were assigned to "Universal stress protein A", 5 to "ArsR-type helix-turn-helix", 5 to "Response to copper ion", 20 to "Virulence", and 7 to "Chaperone". Following the reviewer's suggestion, we have added these gene lists in Table S4.

For the question regarding "cell wall" and "cell wall and cell processes" enrichment, we further checked the genes within these two categories as defined by the different datasets (GO and Mycobrowser). We noticed that these two terms had very different compositions of genes, which explained the discordance that the reviewer has pointed out. After a closer check of the "Cell wall" term in GO database, we noticed that this was not a specific term and that this group included genes from various functional categories. Indeed, GO database advises against using the term "Cell wall" for manual annotation due to the low species-specificity. However, we manually checked other categories and they didn't have this issue. In this revised manuscript, we removed "cell wall" in Fig. 2a. However, respiration and hypoxia refer to two different categories and they only overlap for 3 genes.

- In line 154-155, the authors consider that $r < 0.21$ shows no degree of correlation. However, in many of the correlation analysis in both main figures and supplementary figures (Fig S5, S6, Fig. 6a, b, and f) the authors accept a degree of correlation for even lower r value. This is confusing and contradictory when it comes to interpreting the output of an analysis. I think $r = 0.21$ is still a degree of correlation. For example; the authors think $r = 0.22$ is an apparent correlation for TP and GC% deviation in Fig S4a. Therefore, the sentence in lines 154-155 should be corrected.

Reply: We thank the reviewer for pointing out this careless statement. In this revised manuscript, we have modified the original sentence to *"We observed no or low-level correlations between the top features (Fig. S4c)".* (Lines 182-183)

- Without being sure, is it possible to take the base substitution rate of regulatory regions of genes and compare this to TP? This is because the plasticity of gene expression is assumed to be primarily linked to the regulatory region of the genes.

Reply: We agree that regulatory regions may play important roles in affecting TPs and a recent study has found that the gene expression level somehow can be predicted by regulatory DNA (Vaishnav et al., Nature, 2022). It is reasonable to hypothesize that there could be a relationship (either positive or negative) between the

substitution rate of regulatory DNA and the levels of TPs. Here, we followed the reviewer's suggestion and calculated the mutational events in the 50 bp of DNA upstream of *Mtb* genes. First, we reconstructed the ancestral sequences of the upstream 50bp DNA of 913 genes using sequences from *Mtb*, *Mab*, *Msmeg* and *M. marinum* by *RAxML* software. We then blasted the sequences from *Mtb* to the reconstructed ancestral sequences and identified mutational events in each 50bp DNA sequence to indicate base substitution rate. The results of this analysis did not show any correlation (neither positive nor negative) between mutational events and TP (**Fig. R2**). After discussions, we speculate that perhaps only a few nucleotides in the regulatory DNA sequences are subject to a differential substitution rate, but such effects might be masked by the other “neutral” or non-varying sites. We agree that this is an interesting possibility that warrants more thorough investigation in a future study.

Fig. R2 Substitution rate of the 50bp upstream DNA of 913 *Mtb* genes showed no significant correlation with TP.

• In prokaryotes, the DNA concentration near the replication origin is typically higher than that near the chromosome terminus due to the bidirectional replication process in replicating bacteria. Considering that the dataset used in this study comprises samples from 73 different conditions, presumably including samples from both the exponential and stationary phases, I am curious whether the genomic location, defined as the distance from the replication origin, could be a valuable feature for predicting TP in the SVM model. It might be worthwhile to incorporate this feature into the current model to assess whether its inclusion enhances TP prediction. Furthermore, I also wonder whether gene vulnerability could serve as another useful feature in the prediction model. It might be worth exploring the potential inclusion of gene vulnerability as a feature to improve the accuracy of TP prediction.

Reply: We appreciate these suggestions and agree that the DNA concentration and gene vulnerability can be potential contributing features for TP. In this revision, following the suggestions from both reviewer 2 and reviewer 3, we added 41 new genetic features to our machine learning model, including genomic location (defined as distance to Ori), vulnerability parameters, and others (see the revised methods section) (**Fig. 3a**). This updated model resulted in an increase of R^2 from 0.16 to 0.25 when using genetic features to predict TP (**Fig. S4a**).

Notably, we found that “genomic location (distance to Ori)” turned out to be one of the high-ranking features (ranking 11st) that contribute to TP (**Fig. 3c, Fig. R3a**). However, this contribution was not manifested as a global correlation between “distance to Ori” and TP (**Fig. R3b**), and therefore we sought to explore the roles of local genomic architecture. Interestingly, we found genes near the Ori had significantly lower TP while genes near the Ter exhibited higher TP (**Fig. R3c-d**). The lower TP near the Ori might be advantageous by minimizing conflicts when replication and transcription simultaneously. In contrast, genes located at Ter may have higher TPs because they are subject to transcription instability due to mechanical damage during chromosomal segregation.

‘Vulnerability index’ ranked 82nd out of the 119 total features. However, several features related to gene vulnerability had high rankings. For example, a gene’s maximum fitness cost when being knocked down ranked

15th, followed by the mean gamma parameter value that ranked 16th (Fig. R3a). In this revised manuscript, we have updated the results accordingly.

Fig. R3 (a) Ranking of features pertaining to the vulnerability index and gene's distance to the Ori. **(b)** The distance of a gene from the Ori shows no significant correlation with TP. **(c-d)** TP decreases as genes are closer to Ori **(c)** and increases as the genes are closer to Ter **(d)**.

- The analysis performed on the TP of genes belong to monocistronic and polycistronic mRNAs very much striking. Therefore, I think the Fig. S4d should be lifted to main text. I wonder if the authors noticed that the correlation between TP of genes in the same operon gets lower when the distance between genes increases. For example, in Fig. S4d (if I interpreted the figure correct), the R for the first gene in the operon; 1st to 2nd =0.56, 1st to 3rd=0.45, and 1st to 4th =0.41 and the second gene in the operon; 2nd to 3rd =0.63, 2nd to 4th =0.44. As you can see with those examples as the distance increases the correlation gets lowered. If this is correct, why is it so? Is the location of the genes in operons related to their TP? Could authors test this further? One could check the TPs of genes in different location in the operon to answer this question.

Reply: The reviewer made a good catch by pointing out this trend: the correlation of TP between genes in the same operon gets lower as the distance between the genes increases. In the revised manuscript we have added a heatmap to emphasize this result (new Fig. 4e). Following the reviewer's suggestion, we further investigated large operons with gene number ≥ 10 . Interestingly, we found that as the pair-wise distance between genes in an operon increases, the correlation of their expression levels decreases (new Fig. S5e). The molecular mechanisms underlying this behavior warrant a follow-up study. We thank the reviewer for highlighting this observation.

Fig. 4 (e) Pearson's correlation coefficients of TP between genes in different operonic positions, i.e., the first, the second, the third and the fourth gene of an operon.

Fig. S5 (e) The closer that two genes are to each other in an operon, the more similar are their expression levels. X-axis represents the distance between two operonic genes, e.g., the interval of adjacent genes is 1. Y-axis represents the Pearson's correlation coefficient of gene expression level between two genes. Pearson's correlation coefficient and corresponding p value are presented.

• TP to GC% deviation for *whiB4* and *sigH*, TP to gene length for *whiB1*, *zur*, *sigD* and *Rv1828* were repeated in Fig. 5d and e and in Fig. S5 and Fig. S6. Moreover, while the R values are the same in two places, the p-value changes. Why is this difference in same analysis shown in two different figures?

Reply: Thanks for pointing out this error due to a typo in our codes (p values in Fig 5d-e were calculated by Pearson correlation, which, however, should be Spearman correlation). We have corrected this and updated the p values in this revised manuscript.

• In Fig. S7d, the conclusion made on the functional enrichment analysis is based on the total of 23 genes (if there is no common genes in the two groups) out of 316 genes for *Msm* and 51 genes (if there is no common genes in the two groups) out of 242 genes for *Mab*. I think the conclusion made on the function and TP has no strong evidence. Therefore, I do not see association of TP to any biological function.

Reply: We accept this critique and was curious if this was due to poor gene annotation or functional category assignment. During this revision, we further checked the David database and noticed that they updated the data source in Dec of 2023. We then repeated this enrichment analysis and found a few more functional categories that are significantly enriched (Fig. S8d-e). The enrichment of metal-related functions, which we previously found across all three species, persisted in the updated result. In the revised manuscript, we have updated the supplementary figures accordingly. We agree, however, that because there were many fewer RNA-Seq samples from *Msm* (192) and *Mab* (106) compared to *Mtb* (894), there is less confidence in our TP calculation of gene expression in these two species.

Term (Msm)	FDR	Count
Universal stress protein A	0.0014	7
Iron	0.0027	22
Oxidoreductase	0.0034	50
Metal ion binding	0.0106	26
Stress response	0.0303	5
Membrane	0.0380	58
Peroxidase	0.0993	6

Term (Mab)	FDR	Count
Ferritin-like superfamily	0.0033	9
Oxidoreductase	0.0041	36
Membrane	0.0262	44

Fig. S8 (d-e) Enrichment analysis conducted on the top 5% highest TP genes in *Msm* (d) and *Mab* (e), using the DAVID platform. The significance threshold for enrichment results was set at FDR < 0.1.

- I would like to express my appreciation for the authors' suggestion to incorporate TP into the analysis of differential expression, along with the provision of data in Table S6. However, I believe it would greatly benefit the readers if Table S6 were accompanied by a more detailed description. In lines 298-300, the authors mentioned the 5th and 95th percentiles without specifying the percentile of what exactly, which might require clarification for better comprehension.

Reply: We appreciate this suggestion. We have added the explanations “the 5th and 95th percentiles” of in the updated supplementary table (now **Table S8**) In addition, we have also added a paragraph in the manuscript to discuss the potential usage of TP in future RNA-Seq analysis (lines 371-378).

"Table S8. 95th and 5th expression level for each Mtb genes and their log2 fold changes compared to gene's mean expression level. q5 and q95 refer to the expression values corresponding to the 5th and 95th percentiles in our dataset of 894 samples."

"Alternatively, we propose incorporating TP into RNA-Seq analysis as a normalization factor. In our curated dataset of 894 RNA-Seq samples, we observed that the fold changes of differentially expressed genes (DEGs) were positively correlated with their TP values (Fig. S9a, see Materials and Methods). A potential way to diminish this effect is to divide log2 fold change (logFC) by the gene's TP. By doing this, logFC values of low-TP genes would be divided by smaller numbers than high-TP genes, which would result in an increase of low-TP genes' ranking in DEGs. After this normalization, the positive correlation between "TP-adjusted logFC" and TP was reduced (Fig. S9b), while the values of logFC and TP-adjusted logFC were still highly correlated (Fig. S9c, Pearson's correlation coefficient: 0.93)."

Minor comments:

- What does the X-axis of hspX, rpoB, and lpqM plots in In Fig 1c present?

Reply: The X-axis represents the ranking of 3,891 *Mtb* genes ordered by their expression ranges (*MinMax*). We have added this explanation in the figure legend.

- The figure legend for Fig. S1b is missing. This should be added. The same legend there is ‘©t’ at the end of sentence describing d panel. This should be corrected.

Reply: Thanks for pointing out this error. We have added the figure legend for Fig S1b (now Fig. S1h) and corrected the typo.

- In Fig. S1c, aSD was used instead of adj-SD. To avoid confusion, it should be adj-SD as in Fig1d instead of aSD.

Reply: Thanks again. We have changed 'aSD' to 'Adj-SD'.

- In line 107, the statement about the sample size reduction to 10 ‘genes’, I guess is wrong. Should it be ‘samples’ or ‘conditions’ instead of ‘genes’?

Reply: We truly appreciate the referee's careful reading of the manuscript and have changed 'genes' to 'samples'.

- In line 115, the number of core genes and variable genes should be mentioned and it would be more appropriate to place the reference here (which is now in Materials and Methods).

Reply: In the revised manuscript, we added the number of core genes and variable genes and placed the reference in the main text (lines 144-146).

- The figure legend for Fig. 2b states that ‘The X-axis is presented on a log scale’. Is it correct for Fig. 2b or should it be for Fig. 2a?

Reply: It is for Fig 2b. We have unified the X-axis to a log scale for TP presentation.

- Pearson and Spearman correction coefficient are represented randomly different (R, R2, r) in the figures and text. It should be a consistent representation in overall figures and text.

Reply: We have unified the correction coefficients as 'R' in the revised manuscript

- In Fig S5 and S6 legends, it should be 'Spearman's correlation coefficient' not 'correction'

Reply: We have changed "correction" to 'correlation'.

- In Fig 4c and Fig. S4a the X-axis of the plots show SD of GC% and GC% deviation, respectively. Do these two terms refer to the same? If yes, one only should be used to avoid confusion. If not, they should describe what they mean

Reply: SD of GC% and GC% deviation are different: "SD of GC%" in Fig 4c refers to the GC% variation of genes within a group (TP bin), while "GC% deviation" in Fig S5a refers to the difference between the GC% of a gene and genome-wide GC% (65.6%). Both of these measures show that TP increases with GC% variation. In this revised manuscript, we have added a sentence to explain this difference (lines 192-194):

"We also calculated gene-level GC% deviation from the genome-wide GC% and observed a positive correlation between GC% deviation and TP (Fig. S5a)."

- I did not quite understand why the authors wanted to pick 50 quantiles and SD GC% on ranked bins in Fig. 4c. They should explain.

Reply: In Fig 4b, we noticed that low-TP genes appear to have a constrained GC content (GC%) that is close to 65%, while high-TP genes have GC%'s far from 65%. To further explore this, we grouped Mtb genes into 50 bins based on their TP values, with bin-1 refers to a group of genes with the lowest TP while bin-50 refers to a group of genes with the highest TP. If low-TP genes do have constrained GC%, the variation in their GC% (measured as SD of GC%) should be smaller than high-TP genes, which is what we found. The bins with lower-TP genes had smaller SD's of GC% than bins with high-TP genes. We have added some additional description for Fig 4c in the main text: *"(c) Positive association between the average TP of the genes in a bin (divided in 50 TP bins) and the standard deviation (SD) of the GC% of the genes in that bin."*

- The X-axis in Fig. S5 should be gene length.

Reply: We have changed "width" to "gene length".

- In Fig. S7a, the number 20 should be 200 in X-axis.

Reply: Thanks for this careful check; we have corrected this lapse (now **Fig. S8a**).

- In line 221, they should refer to Fig. 2a instead of Fig. 3a

Reply: Thanks, it was corrected.

- In some cases, gene names were written as protein names such as in Fig. 5d and e. For example; it should be sigD instead of SigD.

Reply: We have now unified gene names to appear in italics and non-capitalized, while regulon names are not in italics but have the first letter capitalized. The panel titles in Fig. 5d and 5e represent the regulon names.

Reviewer #3 (Remarks to the Author):

This paper describes a meta-analysis of hundreds of Mycobacterium tuberculosis RNA-seq datasets to explore the genetic basis of transcriptional plasticity. Overall I like the approach taken, and the manuscript is well laid out. However I have concerns about underlying data quality / pre-processing that may impact conclusions.

Major concerns:

It seems possible that the observed relationships between operon length / gene length / GC and Transcriptional plasticity are due to technical artifacts in the underlying RNA-seq datasets (primarily differences in coverage uniformity or GC bias). These biases would also effect data from different species giving the appearance of conserved patterns of TP (figure 6). Extra analyses are required to rule out this possibility.

Reply: We share the concern from the reviewer that the technical factors, such as coverage uniformity and GC bias, could vary between RNA-Seq datasets generated in different studies. In this revision, we performed additional analyses to address “coverage uniformity” and “GC bias” and presented the results in the following comments. In the revised manuscript, we added a paragraph to describe these analyses (lines 119-123).

“Because other technical factors could potentially affect the estimation of TPs, such as read coverage uniformity and GC content-associated sequencing bias¹⁸, we conducted constrained analyses to assess their influence by controlling for each factor (Fig. S2a-b). TPs calculated from subsets of samples grouped by their degrees of mRNA coverage uniformity or GC% preference were still highly correlated with TPs calculated from the entire dataset (Fig. S2a-b).”

-Does the data normalization approach used sufficiently account for all batch effects? It likely works well for comparison of the same genes across different samples, but I am concerned that there may be biases remaining that undermine comparisons across different genes (e.g. genes of different length or GC content)

Reply: The reviewer asks whether the biases associated with batch effects would affect the comparisons across different genes. In this revision we performed two additional analyses to address this concern. First, we restricted our analysis to each of three individual batches of data (PMIDs: 32348771, 29511081 and 32916109) that were generated from a single study. This subsampling approach can further minimize the batch effects that exist between different studies. We calculated TPs in each individual subset and compared them to TPs calculated from the entire dataset. We found that the TPs estimated from the subsets are highly correlated with TPs calculated from all samples (new **Fig. S2c-d**).

Second, to control for the effect of GC content on TP analysis, we focused on 316 genes with similar GC% (65.3% ~ 65.9%) for genetic features assessment. We found that even within this subset of genes with similar GC%, TP still showed a significant negative correlation with gene length and positive correlation with operon size (**Fig. R4a-b**), consistent with the results obtained analyzing all genes (**Fig. 4a and 4d**).

These two constrained analyses provide confidence that our TP estimation and their relationship to genetic features are robust to potential effects associated with batches or GC content. In the revised manuscript, we have the following sentences to describe the results from this subsampling approach (lines 123-129):

“Additionally, to exclude the possibility that our results were biased by technical factors associated with experimental batches, we applied our analyses to three independent BioProjects with relatively large numbers of samples, and estimated TPs within each of these batches (Fig. S2c). We found that TPs calculated from individual batches, despite their smaller sample sizes, still showed a high correlation with the TPs calculated from the entire dataset (Fig. S2d). Together, these analyses give us confidence that the TP estimation is robust to technical biases associated different experimental batches.”

Fig. R4. The relationship between TP and genetic features in a subset of genes with similar GC% (65.3% ~ 65.9%). **(a)** TP negatively correlates with gene length. **(b)** Genes from polygenic operons have significantly higher TPs than genes from monogenic operons.

-How reliable is the correlation between TP and operon length and gene length? I wonder if this could be an artifact of poor coverage uniformity in some underlying datasets (e.g. where RNA degradation results in failure to cover 3' end of long genes or operons in some datasets). You could check RNA-seq coverage uniformity of datasets (eg by visualization in IGV, or by making metagene using deepTools), then segregate the data based on gene body coverage uniformity metrics. Compare the gene length-TP relationship in conditions exhibiting the most uniform coverage against those with the least uniform coverage.

Reply: We appreciate these insightful comments. In the revised manuscript, we followed the reviewer's suggestion and assessed coverage uniformity by measuring the read coverage of gene body for each sample using the function 'geneBody_coverage.py' from the RSeQC package, which was developed to assess the global distribution of read counts on gene body of an RNA-seq sample (Wang et al., Bioinformatics, 2012). We calculated the coefficient of variation (CV) of read coverage as a measurement of coverage uniformity for each sample, where uniform read coverage is indicated by a low CV. We found that most samples had relatively low CV values (<0.1) (presented as new Fig. S2a), suggesting generally uniform distribution of sequencing reads on a gene body. Next, we equally divided 894 samples into 3 subsets according to the CV of read coverage and recalculated TPs in each subset (groups A, B and C in Fig. S2a). We found that TPs calculated from the three groups were highly correlated with TPs calculated from the entire dataset (Fig. S2a). Moreover, this subsampling analysis also recaptured the relationships between TPs and genetic features, such as gene length and operon size (Fig. R5a-b). Together, these additional analyses obtained through stratifying uniformity indicate that TP estimation is not sensitive to variations in read coverage. In the revised manuscript we included the following sentences in the main text to address coverage uniformity (lines 119-123):

"Because other technical factors could potentially affect the estimation of TPs, such as read coverage uniformity and GC content-associated sequencing bias¹⁸, we conducted constrained analyses to assess their influence by controlling for each factor (Fig. S2a-b). TPs calculated from subsets of samples grouped by their degrees of mRNA coverage uniformity or GC% preference were still highly correlated with TPs calculated from the entire dataset (Fig. S2a-b)."

Fig. R5. TPs calculated from coverage-uniformity subsets showed significant correlation with gene length (a) and operon size (b). The three plots in panel a and b refer to the three groups (A, B and C) in Fig. S2a.

-Similarly, how reliable is the correlation with TP and GC content? Regions of GC are notoriously problematic for library prep and sequencing and are often associated with coverage biases. Again try to assess the GC coverage bias of individual samples and check if TP-GC relationship is robust.

Reply: Thanks for this question. We followed the reviewer's suggestion and evaluated the effect of GC coverage bias on TP-GC relationship. GC coverage bias refers to over- or under-amplification on GC-poor or GC-rich regions during sequencing. As expected, we observed an overall negative correlation between genes' GC% and read counts (710 of the 894 (79.4%) samples had Spearman R lower than 0, new Fig. S2b). However, it was unclear to what extent this correlation would affect our TP estimation. Therefore, we re-calculated TPs in samples from low GC bias group (518 samples, R from -0.1 to 0.1) and samples from relatively high GC bias group (340 samples, $R < -0.1$) (new Fig. S2b). We found that the TPs calculated from both groups were highly correlated with the TPs calculated from the entire dataset (Fig. S2b). In addition, the observation from the full dataset that the TP decreases around the average GC% was also seen in both the high and low GC bias groups (Fig. R6). Therefore, these analyses indicate that TP and its relationship with GC% are robust to variations in GC bias. The newly added sentences cited immediately above, in the response to the referee's concerns about operon and gene length, also address GC biases (lines 119-123).

Fig. R6. The observation that TP decreases around the genome-wide GC% was validated in both low-GC coverage bias (a) and relatively high-GC coverage bias (b) groups.

I like the overall approach used to explore the genetic basis of TP (figure 3), but R^2 of 0.17 seems disappointingly low, and most of the variance in TP remains unexplained. Could the table of features be expanded (e.g. to include specific TFBS annotations / promoter sequence composition / other?). Investigation of some of the regions in figure 1.e. with very high TP could provide additional clues.

Reply: We appreciate this suggestion. After thorough discussions and literature review, we have added 41 additional genetic features to cover: 1) numeric descriptors of the promoter sequences, 2) the location of individual genes on the genome and 3) gene vulnerability. We found that these new features, together with parameter tuning, have substantially improved the predictive performance of our light-GBM regression model, resulting in a significant increase of R^2 from 0.17 to 0.25. We have updated the results accordingly in the revised manuscript.

Minor comments / concerns:

-Introduction, line 24-26. The concept and definition of Transcriptional Plasticity was not clear to me and took several readings. It would be helpful to give a clearer more expansive explanation (e.g. TP can be visualized as the extent to which gene expression levels shift across multiple conditions. For instance, a gene with high TP would exhibit significant changes in its expression levels under different environmental stresses, while a gene with low TP would show relatively stable expression regardless of the environment.)

Reply: Thanks for this suggestion. We have added a schematic plot to visualize the expression changes of high-TP and low-TP genes across different conditions (New **Fig. 1a**). We have also added additional text to provide a clearer description of TP (lines 39-41):

"For instance, a gene with high TP exhibits substantial changes in expression across different conditions, while a gene with low TP maintains relatively stable expression regardless of environmental changes (Fig. 1a)."

-Results, line 89-92. Can you speculate on why *hspX* and *ipqM* might have different TP?

Reply: In the manuscript we use these two genes to illustrate that there are marked differences in the ranges of expression between individual *Mtb* genes, an observation that led us to perform the subsequent analyses to learn about the TP and the genomic features that affect it.

hspX is a member of the hypoxia responsive DosR regulon, which is enriched in high TPs (**Fig. 2a**). while *ipqM* is a lipoprotein peptidase that belongs to "cell wall and cell process" genes that generally exhibit low TPs (**Fig. 2b**). We find that *hspX* is short (435 bp), located in a polygenic operon, targeted by 2 activators, and has a GC% that deviates from the genome-wide GC% (61.6%, ~10th percentile), all of which are associated with high TPs (**Fig. 4h**). On the contrary, low-TP gene *ipqM* is longer (1,497 bp), located in a monogenic operon, has no annotated activators, and has a GC% of 64.9% that is much closer to genome-wide GC% of 65.6%, all of which are associated with low TPs (**Fig. 4h**).

-Fig 1.e. It would be interesting to know what genes are at genomic regions with 'spikes' in extreme TP e.g. at 2, 2.25, 3, 3.5MB. Are these operons? Are they horizontally transferred sequences? Visual inspection of these regions may provide clues about causes that are not explained by the SVM model.

Reply: We appreciate these insightful comments. Below are the zoom-in views of the four TP hotspots (new **Fig. S2g**). These hotspots encode short strings of co-cistronic genes: Rv1736c– Rv1737c (*narX*, *narK2*) at 1.9 Mb, Rv2028c - Rv2031c (Rv2028c, *pfkB*, Rv2030c, *hspX*) at 2.2 MB, Rv3132c - Rv3134c (*devS*, *devR*, Rv3134c) at 3.5 MB, and Rv2623 to Rv2628 at 2.9 MB (**Fig. S2g**, genes in black belong to operons). We found that these genes' homologues are shared by *Msm*, *Mab* and other species in the *Mycobacteria* genus, and thus were unlikely to be acquired by the *Mtb* genome via horizontal transfer. Most of these belong to the DosR hypoxia responding regulon (**Fig. S2g**, blue dots represent genes in DosR regulon). We have included genome location in our machine learning model and have added additional text in lines 135-137:

*"Interestingly, there were a few "spikes" in the *Mtb* genome that exhibited extremely high TP (Fig. 1f), and most of these genes belong to the DosR regulon (Fig. S2g)."*

Fig. S2 (g) A zoom-in view of the four high-TP hotspots in Fig. 1f (top). Black texts represent operonic genes, and blue dots represent genes in DosR regulon (bottom).

-Line 105: “Using a bootstrap approach similar to that described in Fig.1d, we found that the 195 high-TP genes in the top 5% percentile demonstrated consistently high TP even when the sample size was reduced to just 10 genes (Fig. S1d, e)”. Should this sentence be “reduced to just 10 samples”?

Reply: We thank the reviewer for pointing out this typo. We have made this correction (line 134).

-Line 127: ‘We hypothesized that high-TP genes may promote phenotypic diversification that confers a selective advantage in the ability of Mtb to survive in fluctuating environments, and therefore these genes might accumulate mutations more rapidly than the rest of the genome.’ - I don’t follow this. Could it not be argued that we expect these genes to be more conserved?

Reply: We appreciate this thinking and agree that this hypothesis can be generated in different ways. We have changed the text to “...and therefore these genes might exhibit rates of evolution that differ from the rest of the genome”.

REVIEWERS' COMMENTS

Reviewer #1 (Remarks to the Author):

I thank the authors for their thoughtful revisions that have strengthened my understanding of their study on transcriptional plasticity and its utility in informing the interpretation of RNAseq datasets. My concerns from the initial review have been addressed.

Minor note: I could not see any code deposited in the github: https://github.com/ChengBEI-FDU/Transcriptional_Plasticity
Please update prior to publication.

Reviewer #1 (Remarks on code availability):

Code does not appear to be available yet, as of 11th March 2024. The only thing I see in the github is a readme file that gives a paragraph summary of the study.

Reviewer #2 (Remarks to the Author):

All my comments are addressed in the revised version of the manuscript.

Reviewer #2 (Remarks on code availability):

I do not have expertise to review the code.

Reviewer #3 (Remarks to the Author):

The authors have responded satisfactorily to all my concerns. In particular they performed additional analyses to assess potential confounding effects of sequencing coverage biases across studies in TP estimates. They have added supplemental figures that demonstrate that this is likely not a serious problem. They also addressed my other minor comments.

Reviewer #3 (Remarks on code availability):

I had a very quick look at the repo. I was able to clone the repo without a problem, but it looks like it needs updating. The README and code are on different branches which is confusing. Scripts refer to ./data which is not included in the repo, so analyses cannot currently be reproduced easily. Need some description in the readme of how to setup the environment for rerunning analyses. Also, it looks like code was not yet updated to reflect the revised manuscript.

Point-by-point responses to reviewers' comments.

Reviewer #1 (Remarks to the Author):

I thank the authors for their thoughtful revisions that have strengthened my understanding of their study on transcriptional plasticity and its utility in informing the interpretation of RNAseq datasets. My concerns from the initial review have been addressed.

Minor note: I could not see any code deposited in the github: https://github.com/ChengBEI-FDU/Transcriptional_Plasticity

Please update prior to publication.

Reply: We have updated our Github link as follows: https://github.com/ChengBEI-FDU/Transcriptional_Plasticity/tree/main.

Reviewer #1 (Remarks on code availability):

Code does not appear to be available yet, as of 11th March 2024. The only thing I see in the github is a readme file that gives a paragraph summary of the study.

Reply: We have updated our Github link: https://github.com/ChengBEI-FDU/Transcriptional_Plasticity/tree/main.

Reviewer #2 (Remarks to the Author):

All my comments are addressed in the revised version of the manuscript.

Reviewer #2 (Remarks on code availability):

I do not have expertise to review the code.

Reviewer #3 (Remarks to the Author):

The authors have responded satisfactorily to all my concerns. In particular they performed additional analyses to assess potential confounding effects of sequencing coverage biases across studies in TP estimates. They have added supplemental figures that demonstrate that this is likely not a serious problem. They also addressed my other minor comments.

Reviewer #3 (Remarks on code availability):

I had a very quick look at the repo. I was able to clone the repo without a problem, but it looks like it needs updating. The README and code are on different branches which is confusing. Scripts refer to ./data which is not included in the repo, so analyses cannot currently be reproduced easily. Need some description in the readme of how to setup the environment for rerunning analyses. Also, it looks like code was not yet updated to reflect the revised manuscript.

Reply: We have updated our latest code and relevant data on Github: https://github.com/ChengBEI-FDU/Transcriptional_Plasticity/tree/main.